**REPORT**

# A Rab1 interactome illuminates a dual role in autophagy and membrane trafficking

Alexander R. van Vliet[1,2] ⓘ, Alison K. Gillingham[1] ⓘ, Tomos E. Morgan[1] ⓘ, Yohei Ohashi[1] ⓘ, Tom S. Smith[1] ⓘ, Ferdos Abid Ali[1,3] ⓘ, and Sean Munro[1] ⓘ

The small GTPase Rab1 is found in all eukaryotes and acts in both ER-to-Golgi transport and autophagy. Several Rab1 effectors and regulators have been identified, but the mechanisms by which Rab1 orchestrates these distinct processes remain incompletely understood. We apply MitoID, a proximity biotinylation approach, to expand the interactome of human Rab1A and Rab1B. We identify new interactors among known membrane traffic and autophagy machinery, as well as previously uncharacterized proteins. One striking set of interactors are the cargo receptors for selective autophagy, indicating a broader role for Rab1 in autophagy than previously supposed. Two cargo receptor interactions are validated in vitro, with the Rab1-binding site in optineurin being required for mitophagy in vivo. We also find an interaction between Rab1 and the dynein adaptor FHIP2A that can only be detected in the presence of membranes. This explains the recruitment of dynein to the ER-Golgi intermediate compartment and demonstrates that conventional methods can miss a subset of effectors of small GTPases.

## Introduction

The Rab proteins, small GTPases of the Ras superfamily, orchestrate the timing and location of many cellular events through their interaction with specific effectors (Homma et al., 2021; Hutagalung and Novick, 2011; Takai et al., 2001). Despite their name, these proteins lack intrinsic GTPase activity. Instead, specialized proteins control their "on" and "off" states by exchanging GDP for GTP (guanine nucleotide exchange factors, GEFs) or by facilitating GTP hydrolysis (GTPase-activating proteins) (Cherfils and Zeghouf, 2013; Lamber et al., 2019; Muller and Goody, 2018). The nucleotide status controls two aspects of Rab function that allows them to act as spatially localized molecular switches. Firstly, the GDP-bound form is bound by GDI, a cytoplasmic chaperone that masks their C-terminal lipid anchor and extracts the Rab from membranes, whereas the GTP-bound form can remain associated with the specific organelle on which it was activated (Barr, 2013). Secondly, the GTP-bound form binds effectors, and thus the Rabs act as spatial landmarks that direct the recruitment of specific proteins to the specific membranes on which the Rab was activated. Thus, to understand the cellular role of a Rab, it is essential to identify its different effectors and upstream regulators.

The Rabs are the largest family within the Ras superfamily, with over 60 members in humans, and Rab1 is conserved in all eukaryotic phyla and is one of only six Rabs that must have been present in the last common ancestor of all eukaryotes (Klopper et al., 2012). Consistent with this degree of conservation, it plays a vital role in cell function and, along with Rab5, it is one of only two Rab activities that is essential for the viability of human cultured cells (Homma et al., 2019). It is also one of the most enigmatic members of the Rab family in that it acts in two distinct processes, membrane trafficking and autophagy. Rab1 in humans exists as two closely related paralogues, Rab1A and Rab1B, which are both widely expressed and appear to be largely functionally redundant (Homma et al., 2019). The best-studied role of Rab1A/Rab1B and their yeast homolog Ypt1 is in membrane trafficking between the ER and the Golgi, where they play an essential role, with Rab1 predominantly localized to the ER-Golgi intermediate compartment and cis-Golgi where it contributes to vesicle capture and compartment organization (Davis and Ferro-Novick, 2015; Galea et al., 2015; Segev, 1991; Westrate et al., 2020). In addition, genetic studies in both yeast and mammalian cells have shown that Rab1 is required for autophagy (Davis and Ferro-Novick, 2015; Haga and Fukuda, 2025; Zoppino et al., 2010). The importance of Rab1 in autophagic processes is underlined by invading pathogens having evolved various mechanisms to inhibit or modulate Rab1 activity to enhance their survival (Dong et al., 2012; Feng et al., 2018; Mishra et al., 2013). However, the precise role that Rab1 plays in autophagy is not fully understood. A recent screen of all human Rabs showed that four Rabs are required for autophagy (Haga and Fukuda, 2025). Three, Rab2, Rab7, and Rab14, contribute to late stages of autophagosome maturation, while only Rab1 activity is essential for initial autophagosome formation. This finding is consistent with the recent report that Rab1 can recruit

[1]MRC Laboratory of Molecular Biology, Cambridge, UK;  [2]Department of Cellular and Molecular Medicine, KU Leuven, Leuven, Belgium;  [3]School of Biochemistry, University of Bristol, Bristol, UK.

Correspondence to Sean Munro: sean@mrc-lmb.cam.ac.uk;  Alexander R. van Vliet: avanvliet@mrc-lmb.cam.ac.uk.

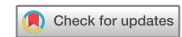

and activate the PI3P-producing VPS34 kinase complex I that plays a key role in the early steps of autophagy (Haga and Fukuda, 2025; Tremel et al., 2021). However, Rabs typically recruit many different effectors, and so it is unclear how else Rab1 might contribute to autophagy. Addressing this question has been challenging, as, unlike dedicated autophagy machinery, Rab1 activity is essential for growth of cultured cells due to its role in membrane traffic.

Previously, our lab used MitoID, a modified proximity biotinylation approach, to identify new interactors for a wide range of Rabs and other small GTPases (Gillingham et al., 2019). Here we apply the MitoID technique to human Rab1A and Rab1B to identify new interactors, including components known to act in membrane trafficking and in autophagy.

## Results and discussion

### Application of MitoID to human Rab1A and Rab1B

Rab1A and Rab1B MitoID constructs were designed as described previously (Gillingham et al., 2019), with the BirA* biotin ligase placed after the C-terminal hypervariable domain of the Rab1 proteins followed by the mitochondrial-targeting transmembrane domain of monoamine oxidase (Fig. 1 A). When expressed in cells, all the Rab1 MitoID constructs were localized to mitochondria and accumulated to similar levels (Fig. S1, A and B). To detect effectors which bind to GTP-bound Rab1, we used mutations known to lock Rab1 and other small GTPases in a GTP-bound form (Q70L for Rab1A, Q67L for Rab1B) or in a GDP-bound form (S25N for Rab1A, S22N for Rab1B) (Feig, 1999; Tisdale et al., 1992). These were compared with a negative control comprising only BirA and the mitochondrial transmembrane domain. HEK293A cells were transiently transfected with plasmids expressing the MitoID constructs, and following a 24-h incubation with biotin, they were lysed, and the biotinylated proteins were isolated with streptavidin and identified by mass spectrometry.

### MitoID with Rab1 efficiently identifies known effectors and regulators

To identify nucleotide-dependent interactors of Rab1A or B, we initially compared levels of biotinylated proteins with those obtained with the negative control to find those enriched for Rab1 binding (Fig. S1 C and Table S1). We then compared the protein levels found with the two different nucleotide states to identify those which are specific for the GTP- or GDP-bound forms. The two comparisons can be evaluated simultaneously on a two-dimensional plot of the fold change of Rab1 vs control plotted against the fold change of GTP vs GDP (Fig. 1, B and C; and Table S1). This approach allows the identification of effectors that bind the GTP-bound Rab by simultaneously evaluating their enrichment against both the control and the GDP-bound Rab. Among the proteins showing the highest enrichment for binding to the GTP forms of both Rab1s were many proteins previously reported to be Rab1 effectors or to be subunits of complexes known to contain at least one subunit that binds directly to Rab1. The known effectors include the Arf GEF GBF1, the actin regulator WHAMM, and the lipid phosphatase OCRL (Hyvola et al.,

2006; Monetta et al., 2007; Russo et al., 2016). The known interacting complexes include the COG complex, which mediates retrograde trafficking within the Golgi apparatus and binds Rab1 via the COG4 subunit, and Vps34 complex I, which generates PI3P on autophagosomes (Tremel et al., 2021; Ungar et al., 2005). Apart from effectors, we also detected a GDP-dependent enrichment of the subunits of the TRAPP complexes that act as GEFs for Rab1 and Rab11 and a GTP-dependent enrichment of the CHM and CHML proteins that present Rabs to the Rab geranyltransferase complex that prenylates them after synthesis (Riedel et al., 2018; Thomas et al., 2018; Zhang, 2003). Taken together, these results indicate that Rab1A and Rab1B remain biologically active and nucleotide-state dependent when used for MitoID. Most of the hits were shared between the Rab1A and Rab1B datasets, with the only obvious difference being the increase in overall hits in the Rab1B dataset (Fig. 1, B and C; and Table S1). Whether this is biologically significant or simply due to differences in levels of active protein is a matter for future studies.

### MitoID with Rab1 identifies potential novel effectors

Among the known Rab1 effectors identified by the MitoID approach were further proteins that have not previously been reported to bind to Rab1. Gene ontology (GO) term analysis of such proteins within the region demarcated by known effectors (Fig. 1, B and C, insets) shows that the most highly enriched terms are those linked to membrane trafficking and Golgi organization, strongly implying that a substantial proportion are also bona fide Rab1 effectors (Fig. 1 D). Two representative proteins that were present in this region for both Rab1A and Rab1B were tested for nucleotide-dependent binding to Rab1. PPP1R37 is a protein phosphatase 1 regulatory subunit of unknown function, and CLEC16A is a GEF for the GTPase Rab2 that acts in both membrane trafficking and autophagy (Gillingham et al., 2019; Haga and Fukuda, 2025; Lorincz et al., 2017; Yin et al., 2017). Both proteins showed GTP-dependent binding to Rab1A by affinity chromatography. PPP1R37 yielded a highly confident AlphaFold prediction for a complex with GTP-bound Rab1 (Fig. 1 E and Fig. S1 D). However, these are just two of 23 proteins found in this region with both Rab1A and Rab1B, which are not known effectors, and so we have focused on validating in depth a subset of the others that seemed particularly striking, with the full list provided in the supplementary material (Table S1). We selected for validation the proteins optineurin (OPTN) and CALCOCO1, as they are major cargo receptors for selective autophagy (Adriaenssens et al., 2022; Lamark and Johansen, 2021). We also selected FHIP2A, a cargo adaptor for dynein.

### Binding of Rab1 to the dynein adaptor FHIP2A requires the presence of a membrane

The presence of FHIP2A in our dataset was intriguing, as a previous study had tested binding between GTP-bound Rab1A and FHIP2A and concluded it could not be detected (Christensen et al., 2021). FHIP2A is a subunit of the FTS–Hook–FHIP (FHF) complex, a key player in cytosolic trafficking that links cargo to dynein motor proteins. It is composed of four subunits: one copy of FTS, two copies of one of three HOOK coiled-coil proteins

Figure 1. **Rab1A and Rab1B MitoID identifies novel Rab1 effectors. (A)** Schematic of the Rab1-MitoID approach in which Rab1 is fused to the promiscuous biotin ligase BirA* and a mitochondrial localization signal from monoamine oxidase (depicted in red), resulting in the relocalization of Rab1 to the mitochondrial membrane. Rab1 interactors will be efficiently biotinylated, while other Golgi proteins and Rab interactors are not. **(B)** Two-dimensional plot comparing the enrichment of proteins biotinylated by Rab1A-MitoID vs control (BirA alone), plotted against the enrichment of proteins biotinylated with GTP-locked Rab1A (QL) vs GDP-locked Rab1A (SN). For the x axis, the value plotted is that for the nucleotide form of Rab1 that gave the greatest fold change over background. For clarity, Rab1A itself is not shown, as it is part of the BirA* construct. Known effectors and regulators are indicated, with known effectors being enriched in the upper right quadrant. Zoomed regions to the right show a region demarcated by well-enriched known effectors with novel proteins of note identified. For all protein identities and enrichment values see Table S1. **(C)** as for (B) except with Rab1B rather than Rab1A. **(D)** Overrepresented GO terms for biological process of the proteins within the regions demarcated by known effectors shown in B and C with the known effectors removed. Ranking is by the summed FDR and enrichment rank. **(E)** Immunoblots of binding to GST-Rab1A–coated beads of the indicated proteins from HEK293A cell lysates. Rab1A was either Q70L (GTP) or S25N (GDP). The known Rab1 effector RABEP1 (rabaptin-5) is included as positive control (Valsdottir et al., 2001). Source data are available for this figure: SourceData F1.

(Hook1, 2, or 3), and a single copy of one of four FHIP proteins (FHIP1A, FHIP1B, FHIP2A, or FHIP2B) (Fig. 2 A). The modularity of the complex underlies its diverse functions, with the subunit combination dictating cargo specificity. For example, FHF complexes containing FHIP1B are linked to Rab5 on endosomes through a direct interaction between the two proteins (Christensen et al., 2021). Although FHIP2A was found to precisely co-localize with Rab1A, the same study found no direct interaction when cell lysates were incubated with GFP-Rab1B–covered beads in the presence of GTP. It was thus speculated that the interaction might be indirect (Christensen et al., 2021).

We hypothesized that the Rab1–FHIP2A interaction is in fact direct, and indeed, AlphaFold 3 confidently predicts a structure for a Rab1A–FHF complex (Fig. 2 B). Consistent with the previous report, assays based on binding to GST-Rab1–coated beads showed no significant interaction between Rab1A or Rab1B and purified FHIP2A (Fig. 2 C). Moreover, binding of FHIP2A to GST-Rab–coated beads does not require the presence of the other subunits of the FHF complex, as a complex comprising FTS, FHIP2A, and the C-terminal region of Hook2 also showed no direct binding (Fig. S2 A). However, AlphaFold predicts that FHIP2A has an N-terminal amphipathic helix adjacent to the Rab1-binding site (Fig. 2, B and D). MitoID labelling of Rab interactors occurs at the mitochondrial membrane, raising the possibility that robust Rab1–FHIP2A binding requires that the GTPase be in a membrane that the amphipathic helix can also bind. To test this, we bound Rab1A to the surface of giant unilamellar vesicles (GUVs), and unlike the result with the Rab1-coated beads, this led to robust recruitment of FHIP2A (Fig. 2 E). Recruitment only occurred when Rab1A was bound to GTP rather than GDP, and removal of the amphipathic helix of FHIP2A abolished binding (Fig. 2 E). We also repeated our MitoID experiments with Rab1A and overexpressed FHIP2A and found FHIP2A to interact robustly with GTP-locked Rab1A, but this was lost with GDP-locked Rab1A or when the amphipathic helix was deleted (Fig. 2 F). Taken together, these results show that Rab1A can bind directly to the FHIP2A subunit of the FHF complex, but that recruitment requires coincident binding to the membrane by the FHIP2A N-terminal amphipathic helix. Almost all the ~160 members of the Ras superfamily of small GTPases are, like Rab1, anchored to membranes via lipid modifications. The conventional methods for identifying effectors for small GTPases are affinity chromatography and yeast two-hybrid screens. These have proven productive, but our findings with Rab1 and FHIP2A raise the possibility that some effectors may have been missed, as their binding is only detectable in the presence of a membrane.

## MitoID reveals a role for Rab1 in recognition of autophagy substrates

The extent of Rab1's role in autophagy is still unclear, and so the presence of the selective autophagy receptors (SARs), TAX1BP1, OPTN, and CALCOCO1, in our MitoID dataset was striking, as they are key components of selective autophagy processes. The SARs function by recognizing ubiquitin and other landmarks on their substrates and simultaneously binding to ATG8 proteins on the phagophore, which elongates to engulf the substrate. We

focused on two SARs, CALCOCO1 and OPTN. CALCOCO1 has been reported to act as a mediator of Golgiphagy (Nthiga et al., 2021), while OPTN has been proposed to have various roles in membrane traffic and Golgi organization but has also been reported to be a key SAR involved in mitophagy (Song et al., 2018; Zhang et al., 2024). Affinity chromatography confirmed direct and nucleotide-dependent binding between Rab1A and both CALCOCO1 and OPTN (Fig. 3, A and B). We also tested binding using GUVs and again observed a strong interaction between the SARs and Rab1A-GTP on the membrane of the GUVs (Fig. 3 C).

To identify the Rab1-binding site on CALCOCO1 and OPTN, we applied cross-linking in combination with AlphaFold structure prediction. AlphaFold 3 predicts that CALCOCO1 forms a homodimer via an extended coiled-coil domain, and Rab1 was confidently predicted to bind to the coil-coil domain close to the N terminus of CALCOCO1 (Fig. 4 A). Chemical cross-linking coupled with mass spectrometry analysis (XL-MS) of the Rab1A–CALCOCO1 complex yielded high-confidence cross-links, providing experimental distance constraints (Fig. 4 B). Most of the cross-links showed Cα–Cα distances consistent with the AlphaFold 3 model, thereby validating the predicted interaction interface (Fig. 4 B). Within this primary binding interface, cross-links mapped interactions involving the switch 2 region of Rab1A with one monomer of the CALCOCO1 homodimer, with residues E234, I237, Q238, S241, E242, K247, E248, and V249 on CALCOCO1 potentially key for Rab1A binding (Fig. 4 C). It should be noted that the predictions and modelling were done with a single copy of Rab1A, but given that the dimeric coiled-coil has rotational symmetry, it is possible that a second copy of Rab1A binds in the same place on the opposite face of the coiled-coil, as is seen in some of the structures of Rab GTPases binding to coiled-coil proteins (Khan and Menetrey, 2013; Pylypenko et al., 2018). Mutating these eight residues in CALCOCO1 abolished Rab1A binding in our GUV-binding assay (Fig. S2 C), without altering protein stability (Fig. S2 B), confirming this interface as the site of Rab1A interaction.

Like CALCOCO1, OPTN exists as a homodimer with elongated coiled-coil domains with functional motifs, including a LC3-interacting region (LIR) between residues 169–209 and a TBK1 kinase-binding site near the N terminus (Phichith et al., 2009; Ryan and Tumbarello, 2018). AlphaFold 3 predicts a well-defined binding interface between Rab1A and the OPTN dimer (Fig. 4 D). Application of XL-MS analysis to the purified Rab1A–OPTN complex strongly supported the protein interaction interface predicted by AlphaFold 3 (Fig. 4 E) and the predicted overall fold of the OPTN coiled-coil domain (Fig. S2 D). The predicted Rab1A–OPTN interface involves contacts with the switch 1 and switch 2 regions of Rab1A, with residues K154, L158, S162, L166, and N169 on OPTN being potentially key residues for Rab1A binding (Fig. 4 F). A previous study reported an interaction between the OPTN and the yeast ortholog of Rab1, Ypt1, and suggested a binding site in between residues 532 and 572 (Song et al., 2018). However, this was determined in the absence of added nucleotide, and given the extensive data presented here, we feel that our work has identified the physiologically relevant effector binding between OPTN and Rab1.

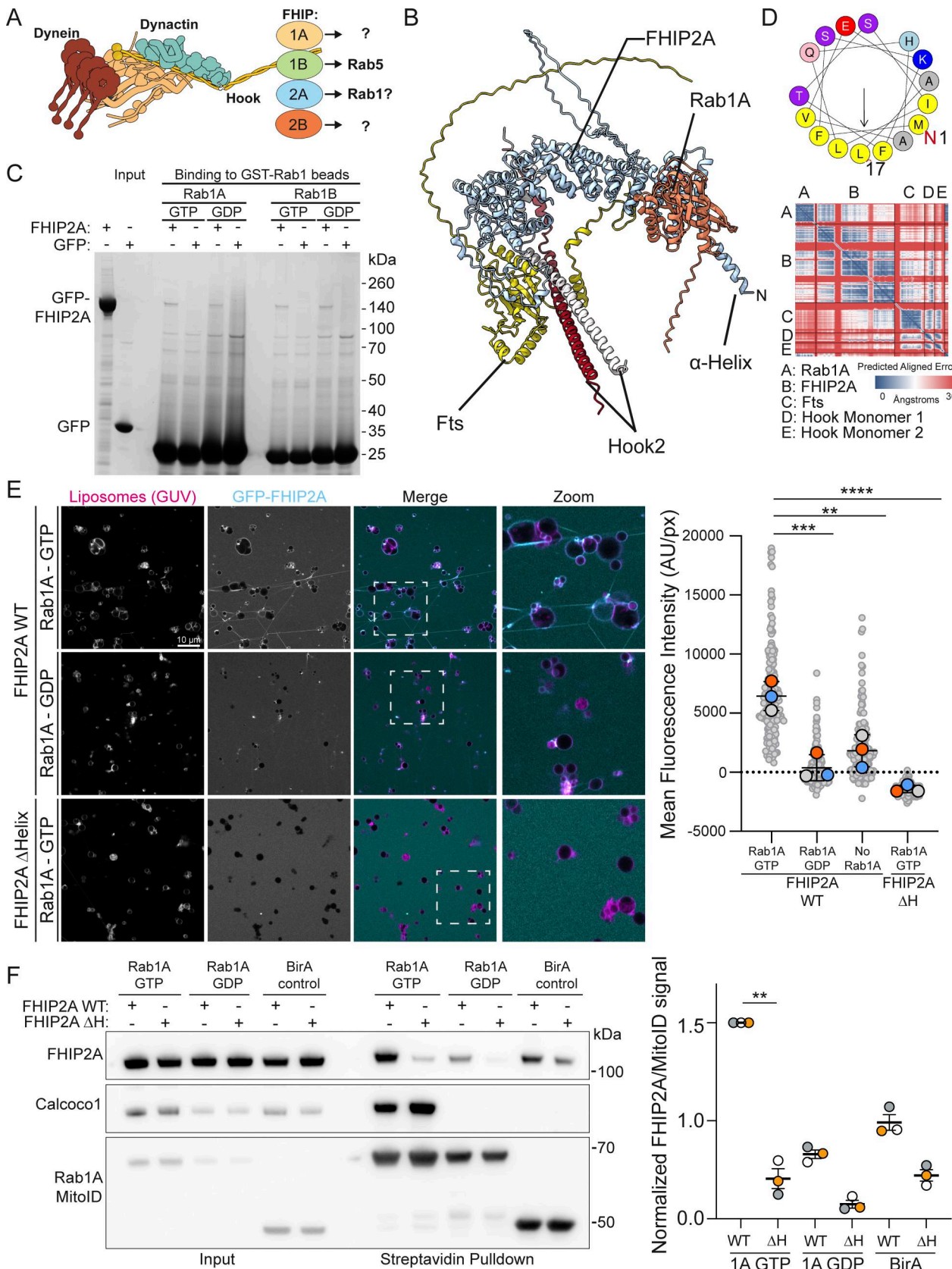

Figure 2. **The interaction between FHIP2A and Rab1A is dependent on Rab1A membrane association. (A)** Schematic of the FHF complex bound to dynein, highlighting the four alternate FHIP subunits. The specific FHIP isoform incorporated into the complex dictates its cargo specificity. **(B)** AlphaFold 3 predicted structure of the Rab1A–FHF complex (consisting of dimeric Hook2 fragments [amino acids 620–719], monomeric full-length Fts, and monomeric full-length

FHIP2A), with accompanying PAE plot. **(C)** Coomassie-stained gel showing an in vitro–binding assay using beads coated with GST-Rab1A or GST-Rab1B and purified GFP-FHIP2A or GFP control. Both GTP- and GDP-locked Rab1A/B proteins were used as indicated. **(D)** Helical wheel plot of the N-terminal 17 residues of FHIP2A with hydrophobic residues (yellow) clustered on one face forming an amphipathic helix. **(E)** GUV-binding assay using GTP- or GDP-locked Rab1A on the GUV, and GFP-FHIP2A. GUVs depicted in magenta and GFP-FHIP2A in cyan. Each large datapoint in the graph depicts the average mean fluorescence intensity of GFP-FHIP2A on a selection of GUV membrane and represents an independent experiment ($n = 3$), with smaller gray datapoints representing all the technical replicates (AU, arbitrary units). The mean ± SD is shown. **P < 0.01; ***P < 0.001; ****P < 0.0001 (one-way ANOVA with Tukey's multiple comparisons test). **(F)** Representative immunoblot of MitoID in HEK293A cells where Rab1A MitoID constructs (detected using anti HA) and 3xFlag-FHIP2A proteins (detected using anti Flag) were transiently expressed. Endogenous CALCOCO1 was used as a positive control. Each datapoint in the graph represents the normalized ratio between the FHIP2A and MitoID construct immunoblot intensities and depicts an independent experiment ($n = 3$). The mean ± SD is indicated. **P < 0.01; (one-way ANOVA with Dunnett's multiple comparisons test). Source data are available for this figure: SourceData F2.

### The Rab1A-binding site in OPTN is important for mitophagy

Having identified the putative Rab1A-binding site on OPTN, we mutated five residues in the binding site to alanine (OPTN Rab1). Mutating these residues did not affect the ability of OPTN to bind to ubiquitin or LC3 (Fig. S3, A and B) and did not affect OPTN dimer formation or stability (Fig. S3 C). In the GUV-binding assay, OPTN Rab1 showed a loss of binding to Rab1A compared with WT OPTN (Fig. 5 A). In the MitoID assay, OPTN Rab1 biotinylation was strongly reduced, indicating a lack of binding to Rab1A in vivo (Fig. 5 B). OPTN is a major cargo receptor for mitophagy; the clearance of damaged mitochondria by autophagy (Wong and Holzbaur, 2014). To test the requirement for the Rab1A–OPTN interaction in mitophagy, we generated stable cell lines expressing WT and OPTN Rab1 in HeLa pentaKO cells that lack five major SARs: OPTN, TAX1BP1, NDP52, NBR1, and p62, causing them to be deficient in selective autophagy pathways, and compared them to HeLa pentaKO cells stably expressing OPTN mutants lacking either ubiquitin (D474N) or LC3 binding (F178S). To measure mitophagy flux, we used a HaloTag attached to a mitochondrial transmembrane protein (Yim et al., 2022). Engulfment of mitochondria by autolysosomes results in degradation of the mitochondrial protein but not the HaloTag bound to a ligand. WT OPTN was able to restore mitophagy to the pentaKO cells, as quantified by the appearance of free HaloTag (Fig. 5 C). However, the OPTN Rab1 mutant deficient in Rab1A binding did not rescue mitophagy to the same extent, and the reduction in mitophagy was similar to that observed with the LIR mutant that is unable to bind LC3. This demonstrates that the Rab1A-binding site is important for OPTN to function in selective autophagy.

### Conclusions

By obtaining a highly specific Rab1 interactome, we have identified further Rab1 effectors that help explain its role in both ER-to-Golgi trafficking and autophagy initiation. Previous work has characterized the role of the FHIP subunits of the FHF complex and the mechanism through which they link dynein to different cargoes (Christensen et al., 2021). We show here that efficient binding of FHIP2A to Rab1 requires the presence of a membrane to which FHIP2A binds through an N-terminal amphipathic helix. Interestingly, in both the in vitro assays and the in vivo cellular MitoID experiments, FHIP2A displayed basal membrane-binding activity that depended on the amphipathic helix, regardless of the presence of Rab1. This dual mode of binding may allow the FHF-dynein complex to associate weakly to membranes to allow scanning for the presence of Rab1, either for the initial recruitment step or if it disengages transiently from Rab1.

Alternatively, the amphipathic helix may direct recruitment to a subset of membranes of a particular lipid composition or curvature. An amphipathic helix is also present at the N terminus of FHIP2B, although its localization and role are currently unknown.

Rab1's role in autophagy has not been investigated in depth in mammalian cells, with Rab1 being essential for viability in contrast to most autophagy machinery. In yeast, the Rab1 ortholog Ypt1 has been reported to bind to the kinases Atg1 and Hrr25, but our MitoID approach did not identify their mammalian orthologues ULK1 and CSNK1D as hits and so was not informative of whether they are Rab1 effectors (Wang et al., 2013; Wang et al., 2015). However, mammalian Rab1 is required for the recruitment and activation of VPS34 kinase complex I on membranes (Tremel et al., 2021), and indeed, the subunits of this VPS34 kinase are strongly enriched in our MitoID interactome. This implies that Rab1 directs the VPS34 kinase complex I to generate the PI3P that recruits downstream autophagy effectors (Nascimbeni et al., 2017). Since PI3P production must happen very early in autophagosome biogenesis, this implies that Rab1 acts at the earliest stages of autophagy. This makes our identification of the major SAR's as Rab1 effectors intriguing, as it is not entirely clear how the autophagosomal membrane source is recruited to the autophagic cargo. The best-characterized interaction is binding to the LC3 proteins on the autophagosome via LIR motifs, but the LC3 proteins are recruited by lipidation that occurs downstream of PI3P production, and so they will not be present at the earliest stages of autophagosome formation. Direct binding of Rab1 to SARs like OPTN provides another mechanism to anchor the autophagosome to cargo. OPTN has also been reported to bind the related GTPases Rab8A, Rab8B, and Rab10 in its role in membrane trafficking, but deletion of all three does not affect autophagy, and so it seems likely that it is the binding to Rab1 rather than Rab8 that is relevant to the autophagy (Okatsu et al., 2025; Zhang et al., 2024). We propose that the binding of Rab1 to OPTN tethers cargo at the earliest stages of autophagosome formation, with OPTN then being handed over to bind to LC3 once the latter has been conjugated to the autophagosomal membrane. This would ensure that clearance of potentially toxic autophagy targets is as rapid and efficient as possible.

In summary, we have identified specific Rab1 effectors that help explain Rab1's essential role in both membrane trafficking and autophagy, as well as further potential effectors whose binding and roles will need to be validated. We have characterized the molecular mechanism of Rab1-FHF complex binding, describing the mechanism through which Rab1 links cargo with dynein motors. In addition, we have characterized a key interaction

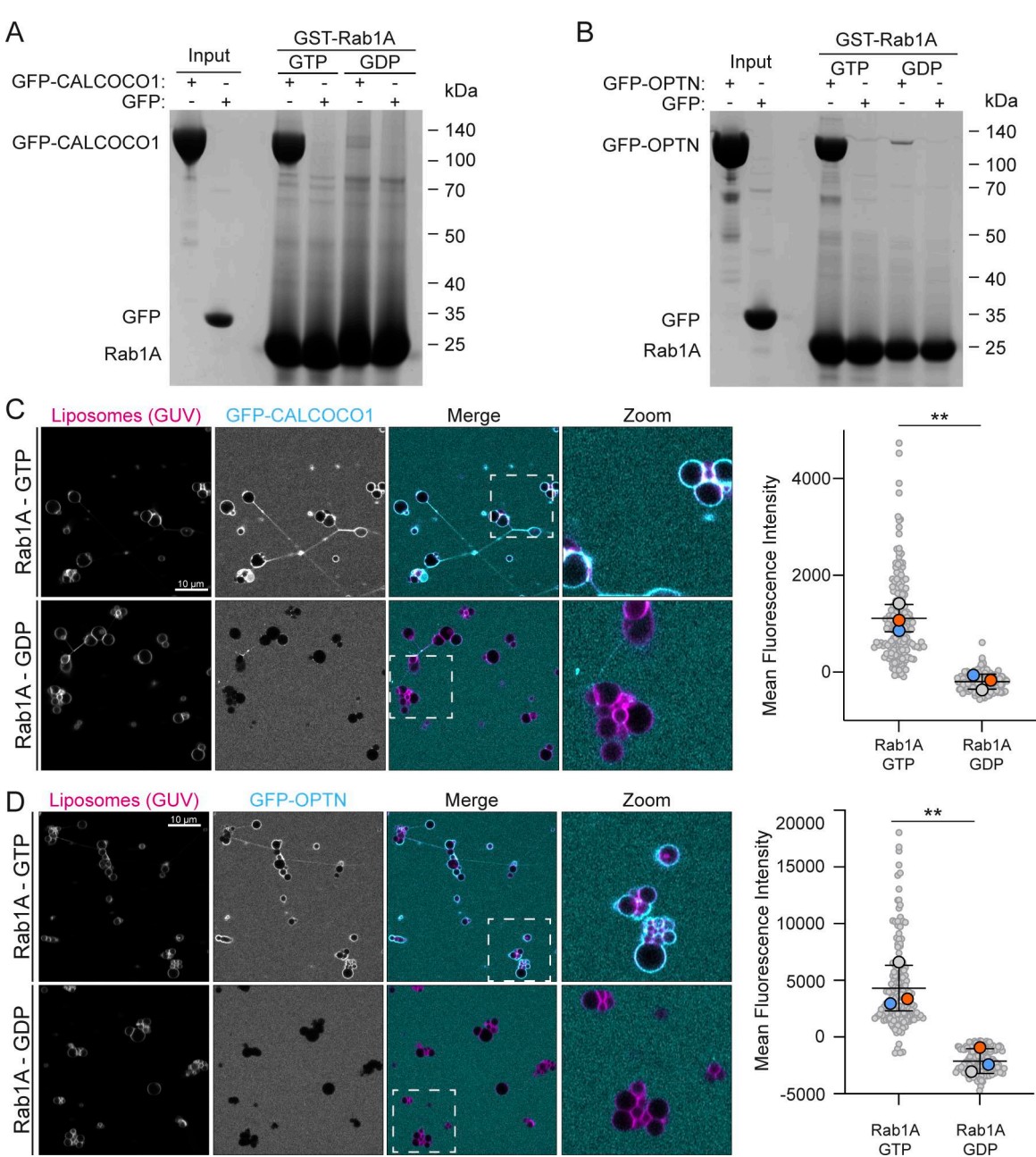

Figure 3. **Rab1A-GTP binds directly to CALCOCO1 and OPTN. (A and B)** Coomassie gels showing in vitro binding to GST-Rab1A–coated beads of either purified GFP-CALCOCO1 (A) or GFP-OPTN (B), with GFP as a negative control. Rab1A was in GTP- or GDP-locked forms as indicated. **(C and D)** GUV-binding assay using GTP- or GDP-locked Rab1A on the GUV with applied GFP-CALCOCO1 (C) or GFP-OPTN (D). Each large datapoint in the graph depicts the average mean fluorescence intensity of GFP-CALCOCO1 or GFP-OPTN on a selection of GUV membrane and represents an independent experiment ($n$ = 3), with smaller gray datapoints representing all the technical replicates (AU, arbitrary units). The mean ± SD is indicated. **P < 0.01 (unpaired $t$ test). Source data are available for this figure: SourceData F3.

between Rab1 and SARs, most prominently the mitophagy SAR OPTN. Disruption of the interaction between Rab1 and OPTN abrogated mitophagy and points to a key role for Rab1 in mitophagy and potentially in selective autophagy in general.

## Materials and methods
### Antibodies
Antibodies used in this study were RABEP1 (610676; BD Transduction Laboratories, RRID: AB_398003), PPP1R37 (HPA041500;

Atlas Antibodies, RRID: AB_10795122), CLEC16A (26257-1-AP; Proteintech, RRID: AB_2880449), Rab1A (13075; Cell Signaling Technologies, RRID: AB_2665537), HA (3F10; Roche, RRID: AB_2314622), TOM20 (ab56783; Abcam, RRID: AB_945896), Flag M2 (F1804; Sigma-Aldrich, RRID), CALCOCO1 (HPA038313; Atlas Antibodies, RRID: AB_10675794), GAPDH (60004-1; Proteintech, RRID: AB_2107436), OPTN (70928; Cell Signaling Technologies, RRID: AB_3073769), and α-tubulin (YL1/2, RRID: AB_305328). All primary antibodies for western blot were used at 1:1,000, and for immunofluorescence at 1:500.

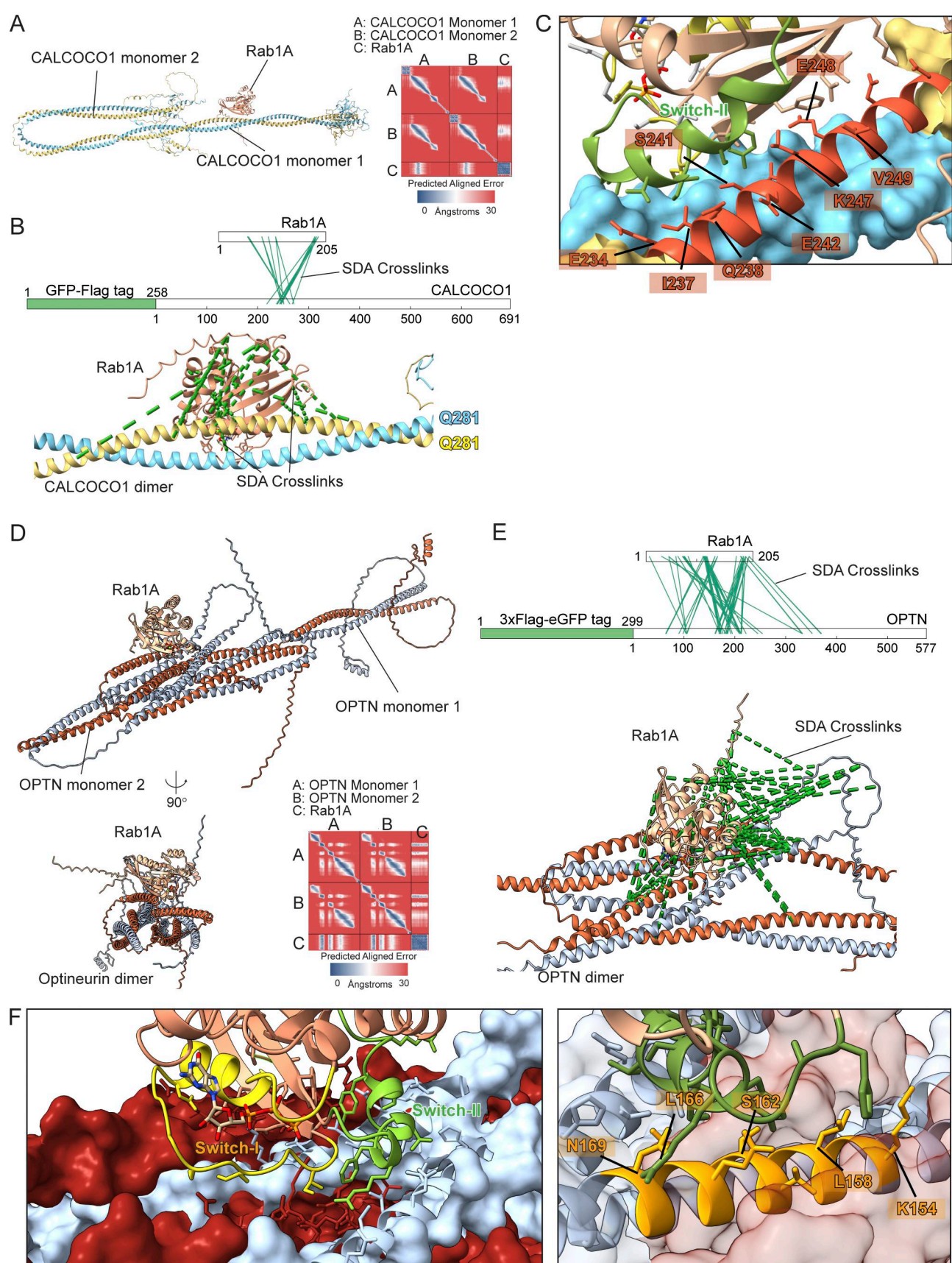

Figure 4. **Molecular characterization of the Rab1A-binding site on CALCOCO1 and OPTN. (A)** Structure of the Rab1A:GTP:Mg²⁺:CALCOCO1 complex formed by two copies of CALCOCO1 and one copy of Rab1A as predicted by AlphaFold 3, with accompanying PAE plot. **(B)** Sulfo-NHS-diazirine (SDA)

interprotein cross-links (green dashed lines) mapped onto the predicted structure of the Rab1A–CALCOCO1 complex. **(C)** The contact interface between Rab1A and the CALCOCO1 dimer, showing ChimeraX calculated contact residues of CALCOCO1 mediating the binding interface (red labels). **(D)** Structure of the Rab1A:GTP:Mg$^{2+}$:OPTN complex formed by two copies of OPTN and one copy of Rab1A as predicted by AlphaFold 3, with accompanying PAE plot. **(E)** SDA interprotein cross-links (green dashed lines) mapped onto the predicted structure of the Rab1A–OPTN complex. **(F)** Magnified image of the contact interface between Rab1A and the OPTN dimer, showing ChimeraX calculated contact residues of OPTN mediating the binding interface (orange labels).

Secondary antibodies used in this study were sheep anti-mouse, donkey anti-rabbit, and goat anti-rat–conjugated HRP antibodies (Cytiva [NA931, NA934, and NA935, respectively], each used at 1:5,000) and goat anti-rat and anti-rabbit secondary antibodies conjugated to Alexa Fluor 488 and 633, respectively (Thermo Fisher Scientific [A-11008, RRID: AB_143165] and [A-21070, RRID: AB_2535731], each used at 1:1,000).

## Plasmids

The MitoID constructs were designed as described previously (Gillingham et al., 2019). Rab1A and Rab1B were engineered to be constitutively active or inactive and lack C-terminal cysteine residues. These sequences were inserted into pcDNA3.1+ (Clontech), followed by a GAGA linker, the coding sequence for the BirA* ligase, a GAGAGA linker, an HA epitope tag, and a mitochondrial-targeting sequence (residues 481–527 of human monoamine oxidase). The 3xFlag C1 plasmid was generated from pEGFP-C1 (Clontech), as previously described (van Vliet et al., 2022), and genes cloned in to be N-terminally tagged with the 3xFlag and linker sequence. Briefly, the GFP coding sequence was excised using AgeI and BamHI restriction sites, and a fragment encoding a 3xFlag tag followed by a linker sequence (5′-GSGAGAGAGAILNSRV-3′) and the original GFP C1 multiple cloning site was inserted using primers 5′-CGCTAGCGCTACCGGTCGCCACCATGG-3′ and 3′-TAGATCCGGTGGATCCCGGGCCCGCGG-5′. GFP-OPTN was cloned into 3xFlag C1 using the XhoI and EcoRI sites with a GFP-OPTN containing plasmid used as a template and using primers 5′-GGACTCAGATCTCGAATGGTGAGCAAGGGCGAG-3′ and 3′-GTCGACTGCAGAATTTTAAATGATGCAATCCATCACGTGAATCTG-5′, resulting in a 3xFlag-(linker)-GFP-OPTN expression construct. To generate 3xFlag-OPTN, the coding sequence for OPTN was cloned into the 3xFlag C1 plasmid using XhoI and EcoRI and primers 5′-GGACTCAGATCTCGAATGTCCCATCAACCTCTCAGC-3′ and 3′- GTCGACTGCAGAATTTTAAATGATGCAATCCATCACGTGAATCTG-5′. To generate 3xFlag-FHIP2A, the coding sequence was subcloned into the 3xFlag C1 plasmid using the same restriction sites and using primers 5′- GGACTCAGATCTCGAATGTTCTCTAAGTTCACTTCTATTCTGCAACACG-3′ and 5′-GTCGACTGCAGAATTTTAGGGAGTGGAAGAGGCATGGTACTTCACG-3′. For GFP-3xFlag-FHIP2A, the previously generated 3xFlag-FHIP2A coding sequence was inserted into the eGFP C1 plasmid using XhoI and EcoRI sites and primers 5′-GGACTCAGATCTCGAATGTTCTCTAAGTTCACTTCTATTCTGCAACACG-3′ and 5′-GTCGACTGCAGAATTTTAGGGAGTGGAAGAGGCATGGTACTTCACG-3′. Plasmids used to create stable cells were generated by cloning the OPTN coding sequence into a retroviral expression plasmid M6P between HindIII and NotI sites using primers 5′-GAAGCTATAGAAGCTTGCCACCATGTCCCATCAACCTCTCAGCTG-3′ and 5′- GGGAGAGGGGCGGCCTTAAATGATGCAATCCATCACGTGAATCTG-3′. The GFP-Flag-CALCOCO1 plasmid was generated using Gibson assembly.

Flag-CALCOCO1 was cloned into a pcDNA3.1(+) vector with an N-terminal GFP sequence using insert primers 5′-ACGACGATAAGAGCGGCCGCGAAGAATCACCACTAAGCCGG-3′ and 5′-AGCCTCCCCATCTCCCGGGTCACTCAAAGGT-3′ and vector primers 5′-ACCCCTTCACCTTTGAGTGACCCGGGAGATGGGG-3′ and 5′-CGGCTTAGTGGTGATTCTTCGCGGCCGCTCTTATCGT-3′. Point mutations in OPTN and CALCOCO1 and deletions in FHIP2A were generated using the Q5 Site-Directed Mutagenesis Kit (E0554S; New England Biolabs) using primers 5′-GCTCCAAGCCTGCCTCTC-3′ and 5′-CATTCGAGATCTGAGTCCGG-3′ (FHIP2A ΔHelix), 5′-GCTGAACTGCAGGCCAAGCTGGCCTCCAGCGGCTCCTCAGAA-3′ and 5′-CACGATGCCCGCCAGGTCTGCCGCCTCTGCTTGTAGCCTCACC-3′ (OPTN Rab1), 5′-GCGAAAGTGCTGACGGCGGCAGCGGAGCTGGACAGGCTTAGAG-3′ and 5′-AGCGATGGTCGCGGCGTCATCCGCTAGCTCCAGGATGCGTGC-3′ (CALCOCO1 Rab1), 5′-AGAAGATTCCtctGTTGAAATTAGGATGG-3′ and 5′-GAGGAGCCGCTGGAG-3′ (OPTN LIR), and 5′-TTACTGTTCTAACTTTCATGCTG-3′ and 5′-ACTTCCATCTGAGCC-3′ (OPTN Ub). GST-Rab1A and GST-Rab1A-6xHIS constructs were generated by subcloning the Rab1A or Rab1A-6xHIS coding sequence into pGEX-6P-2 using BamHI and XhoI.

## Cell lines

Cell lines used in this study were HEK293A cells (CVCL_6910), HEK293T cells (CVCL_0063), HeLa cells (CVCL_0030), HeLa 5KO cells (Lazarou et al., 2015), Expi293 cells (Thermo Fisher Scientific), and Sf9 cells (CVCL_0549). Cells were regularly checked to confirm that they were mycoplasma free using MycoAlert (Lonza).

## Western blotting

For western blotting, cells were lysed on ice using either a modified TNTE buffer (20 mM Tris, pH 7.4, 150 mM NaCl, 1% wt/vol Triton X-100, 5 mM EDTA, 0.5 mM TCEP, and 5% glycerol) or LMNG buffer (20 mM Tris, pH 7.4, 150 mM NaCl, 1% wt/vol lauryl maltose neopentyl glycol [LMNG], 0.5 mM TCEP, and 5 mM EDTA). Each buffer was supplemented with cOmplete EDTA-free protease inhibitor tablets (04693116001; Sigma-Aldrich). Following lysis, debris was pelleted by centrifugation at 16,000 × g. The supernatants were then resolved by electrophoresis on NuPAGE Bis-Tris 4–12% gels (Life Technologies), after which proteins were transferred to a PVDF membrane (Millipore) for immunodetection. Membranes were blocked in 5% (wt/vol) milk in PBS with 0.1% (vol/vol) Tween-20 (PBS-T) for at least 30 min and incubated ON at 4°C with primary antibody diluted in the same solution. Membranes were washed three times in PBS-T before incubation with HRP-conjugated secondary antibodies in 0.1% (wt/vol) milk in PBS-T for 2 h, washed three times in PBS-T, and developed with Luminata Crescendo Western HRP substrate (Merck, WBLUR) using the Bio-Rad ChemiDoc MP using the chemiluminescence detection mode and quantified with Image Lab (Bio-Rad).

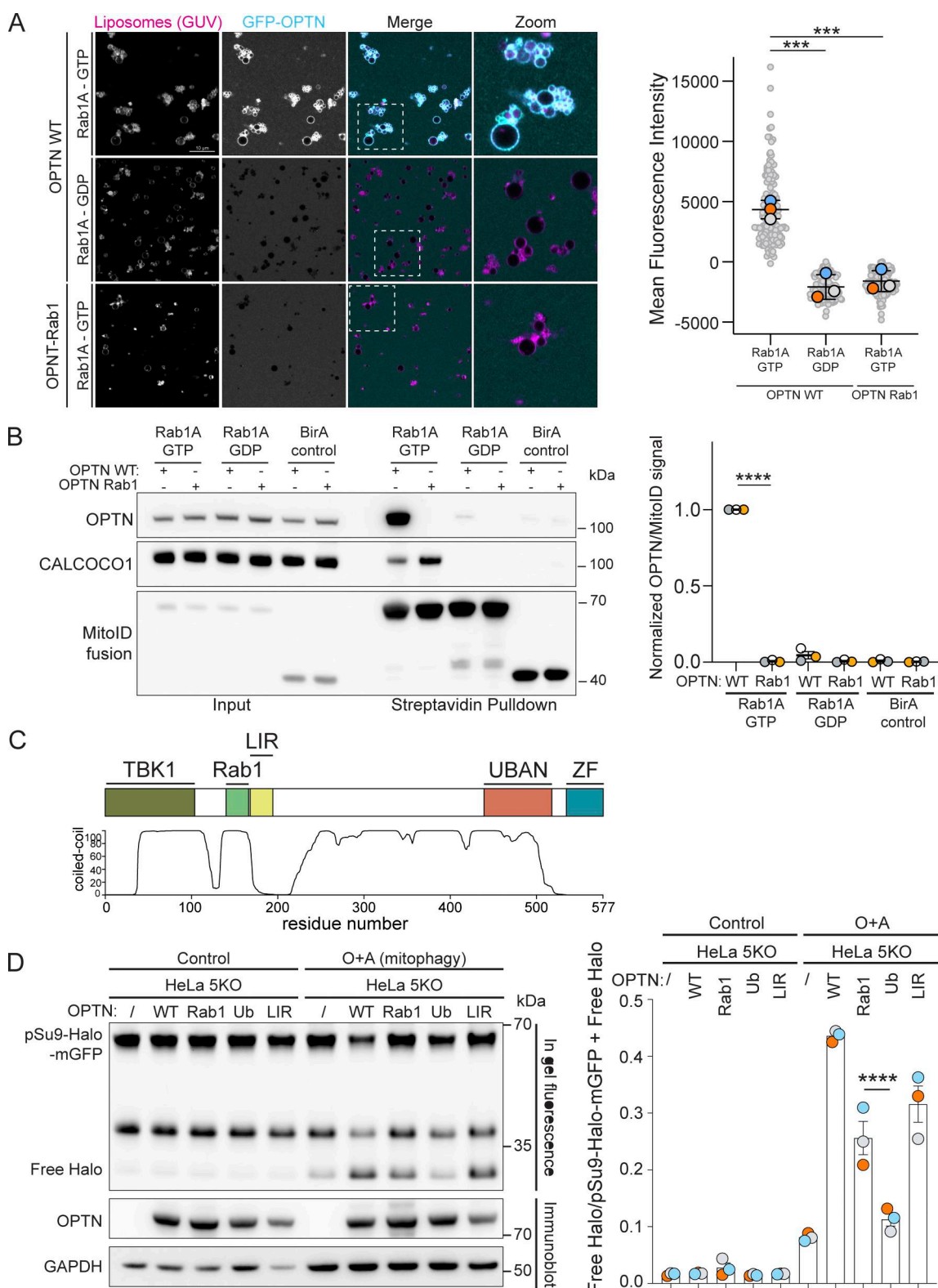

Figure 5.   **The Rab1A–OPTN interaction is required for mitophagy. (A)** GUV-binding assay using GTP- or GDP-locked Rab1A bound to GUVs before applying GFP-OPTN WT or Rab1-binding mutant. Each large datapoint in the graph depicts the average mean fluorescence intensity of GFP-OPTN WT or Rab1-binding mutant on a selection of GUV membrane and represents an independent experiment (*n* = 3), with smaller gray datapoints representing all the technical replicates (AU, arbitrary units). The mean ± SD is indicated. \*\*\*P < 0.001 (one-way ANOVA with Tukey's multiple comparisons test). **(B)** Representative immunoblot of Rab1 interactors following MitoID in HEK293A cells where Rab1A MitoID constructs (detected using anti HA) and 3xFlag-OPTN proteins (detected using anti Flag) were transiently expressed. Endogenous CALCOCO1 was used as a positive control. Each datapoint in the graph represents the normalized ratio between the OPTN and MitoID construct immunoblot intensities and depicts an independent experiment (*n* = 3). The mean ± SD is indicated.

****P < 0.0001; (one-way ANOVA with Dunnett's multiple comparisons test). **(C)** Schematic of the overall structure of OPTN, with different binding regions annotated (TBK1, TBK1-binding domain; Rab1, the Rab1-binding domain; LIR, LC3 interacting-region; UBAN, ubiquitin-binding domain in ABIN proteins and NEMO; ZF: zinc finger domain). The predicted coiled-coil regions of OPTN are also illustrated as predicted by MARCOIL (Delorenzi and Speed, 2002). **(D)** Representative immunoblot and in-gel fluorescence analysis of cell lysates of pentaKO HeLa cells stably expressing pSu9-HaloTag-mGFP and Parkin and either mock transfected (/) or expressing OPTN WT, or the mutant constructs OPTN Rab1, OPTN Ub (unable to bind ubiquitin), or OPTN LIR (unable to bind ATG8/LC3 family proteins). To assay mitophagy, cells were pulse-labelled with 100 nM TMR HaloTag ligand and incubated in medium containing 1 μM oligomycin and 5 μM antimycin for 24 h to induce mitophagy (O + A). Each datapoint in the bar graph is an independent experiment representing the normalized ratio between the free HaloTag and the combined pSu9-HaloTag-mGFP + free HaloTag fluorescence intensities (*n* = 3). The mean ± SD is indicated. ****P < 0.001 (one-way ANOVA with Tukey's multiple comparisons test). Source data are available for this figure: SourceData F5.

## Affinity capture of biotinylated protein

HEK293A cells were grown in two 15-cm culture dishes to ~75% confluence and transfected with 10 μg plasmid and 24 μl Lipofectamine 2000 (11668027; Invitrogen) in 1 ml Opti-MEM (31985070; Thermo Fisher Scientific) according to the manufacturer's instructions. One day after transfection, biotin (B4501-100MG; Sigma-Aldrich) was added to a 50 μM final concentration, and cells were incubated for a further 18 h at 37°C. Cells were scraped in ice-cold PBS and pelleted by centrifugation (1,000 × *g*, 5 min), resuspended in LMNG buffer (20 mM Tris, pH 7.4, 150 mM NaCl, 1% wt/vol LMNG, 0.5 mM TCEP, 5 mM EDTA, and 1× cOmplete protease inhibitor cocktail tablet), and incubated for 15 min at 4°C with periodic vortexing. After centrifugation at 16,100 × *g* for 10 min at 4°C, the supernatants were added to 500 μl Dynabeads MyOne Streptavidin C1 beads (10099482; Invitrogen) that had been pre-washed twice in the same buffer. The beads were incubated at 4°C ON with rotation, washed twice in Wash Buffer 1 (2% SDS PAGE and cOmplete inhibitors), three times in Wash Buffer 2 (1% [vol/vol] Triton X-100, 0.1% [wt/vol] deoxycholate, 500 mM NaCl, 1 mM EDTA, 50 mM HEPES, and cOmplete inhibitors, pH 7.5), and three times in Wash Buffer 3 (50 mM Tris, pH 7.4, 50 mM NaCl, and cOmplete inhibitors). For western blot analysis, the beads were incubated in 75 μl SDS sample buffer containing 3 mM biotin at 98°C for 5 min to release the biotinylated proteins. For mass spectrometry, beads were resuspended in 50 μl Wash Buffer 3 and frozen at –80°C until analysis. All MitoID experiments were performed with biological replicates: the three experiments that constitute a triplicate set for a given GTPase performed on cells transfected independently on different days and processed separately.

## Mass spectrometry

For mass spectrometry analysis, bound protein was eluted by a 10-min incubation at 95°C of the streptavidin beads in presence of 200 μl of 5% SDS with 4.5 mM biotin. Cysteine reduction was performed by a 10-min incubation at 60°C of the eluates with 4 mM final concentration of DTT. Subsequent alkylation was with a 45-min incubation at RT in the dark with 14 mM final concentration of iodoacetamide. Protein aggregation capture–based digestion was performed on the KingFisher Apex magnetic particle processor (Thermo Fisher Scientific). Reduced and alkylated protein eluates were combined with 25 μl of MagReSyn Hydroxyl magnetic beads (Resyn Biosciences). Aggregation was induced by the addition of 800 μl of acetonitrile (final concentration ~70%). Beads were washed three times with 100% acetonitrile, followed by two washes with 70% ethanol to remove contaminants. Proteins were digested on-bead ON at 37°C in

25 mM ammonium bicarbonate containing 0.2% RapiGest (Waters, wt/vol) using 2 μg of trypsin. After digestion, samples were acidified with formic acid to pH <3, incubated at 37°C for 45 min, and the degradation by-products of RapiGest were removed at 13,000 rpm for 15 min, and supernatants were desalted using cation exchange tips. Samples were analyzed using a Thermo Ultimate 3000 (Thermo Fisher Scientific) hyphenated to a Thermo QExactive HFX (Thermo Fisher Scientific). Approximately 1 μg of protein was loaded on a trapping column (Thermo Fisher Scientific, PepMap100, C18, 300 μm × 5 mm) and resolved on the analytical column (Aurora Ultimate XT 25 cm C18 from IonOpticks) at a flow rate of 300 nl/min using a gradient of 97% A (0.1% formic acid) and 8% B (80% acetonitrile 0.1% formic acid) to 25% B over 35 min, then to 40% B for additional 10 min. Data-independent analysis was carried out using 8 staggered windows of 42 Th in width over 400–700 m/z mass range. MS2 DIA windows were acquired at a resolution of 60 k (max IT of 106 ms, AGC target of 3e6). The full MS scan was acquired at 60 k resolution with a Max IT of 60 ms (1e6 AGC target) using a scan range of 400–1650 m/z. Raw data were imported, and data were processed in Proteome Discoverer v3.1 (Thermo Fisher Scientific). The raw files were submitted to a database search using Chimerys against the *Homo sapiens* UniProt database (UP000005640_9606; https://www.uniprot.org/proteomes/UP000005640; UniProt, 2025). Common contaminant proteins (human keratins, BSA, and porcine trypsin) were added to the database. The spectra identification was performed with the following parameters: up to two missed cleavage sites allowed, fixed modification of carbamidomethylation of cysteine, and oxidation of methionine as variable modifications. Only rank 1 peptide identifications of high confidence (FDR < 1%) were accepted. The proteomics data have been deposited to the ProteomeXchange Consortium via the PRIDE partner repository (Perez-Riverol et al., 2025), with the dataset identifier PXD065220 (https://proteomecentral.proteomexchange.org/ui?pxid=PXD065220; ProteomeXchange, 2025a).

## Proteomics data analysis

The peptide-level output from Proteome Discoverer was processed and filtered using the QFeatures (v1.18.0) and biomasslmb R packages. Peptides were filtered to remove common contaminants, nonunique master proteins, or proteins with a single peptide. Peptide intensities were then log2-transformed and normalized using the "diff.median" normalization method with the QFeatures::normalize function. Peptides with >15 missing values (out of 18 samples) were removed before summarizing to protein-level abundances using the MsCoreUtils::

robustSummary function. Proteins with a false discovery rate >1% were then discarded. To limit the impact of imputation, a restricted imputation procedure was employed. Where <2 out of 3 replicates for a given experimental group were quantified, missing values were imputed using the QFeatures::impute function and the "MinProb" method. Proteins with imputed values in both conditions being compared were not subjected to statistical testing. Statistical testing was performed using the limma R package. Specifically, the treat function was used, with the following arguments: lfc = 1 (null hypothesis set as log$_2$-fold-change < 1); trend = TRUE; robust = TRUE. P values were corrected for multiple testing using the Benjamini–Hochberg FDR procedure (Benjamini, 1995). Proteins with adjusted P values < 0.05 were deemed to have significantly different abundances between the conditions. GO overrepresentation analysis was performed against all human proteins as background using ShinyGO 0.82 with an FDR cutoff of 0.05 and an nGenes size minimum of 3 (Ge et al., 2020).

## Cross-linking mass spectrometry

Complexes were mixed with sulfo-SDA (2 mM) and incubated on ice for 5 min before being cross-linked for 10 s with 365 nm UV radiation from a home build UV LED setup. Cross-linking reactions were quenched with the addition of Tris-HCl to a final concentration of 50 mM. The quenched solution was reduced with 5 mM DTT and alkylated with 20 mM iodoacetamide. SP3 protocol as described by Batth et al. (2019) and Hughes et al. (2019) was used to cleanup and buffer exchange the reduced and alkylated protein: in brief, proteins were washed with ethanol using magnetic beads for protein capture and binding. The proteins were resuspended in 100 mM NH$_4$HCO$_3$ and digested ON at 37°C with trypsin (Promega) at an enzyme-to-substrate ratio of 1:20 and protease max 0.1% (Promega). Cleanup of digests was with HyperSep SpinTip P-20 (Thermo Fisher Scientific) C18 columns, using 60% acetonitrile as the elution solvent, and the peptides were dried in a Speed Vac Plus (Savant).

Dried peptides were resuspended in 30% acetonitrile and fractionated by size-exclusion chromatography on a Superdex 30 Increase 3.2/300 column (GE Healthcare) at a flow rate of 20 µl/min using 30% (vol/vol) ACN 0.1 % (vol/vol) TFA as a mobile phase. Fractions were taken every 5 min, and the fractions containing cross-linked peptides (2–7) were collected. Dried peptides were suspended in 3% (vol/vol) acetonitrile and 0.1 % (vol/vol) formic acid and analyzed by nanoscale capillary LC-MS/MS using an Ultimate U3000 HPLC (Thermo Fisher Scientific) to deliver a flow of 300 nl/min. Peptides were trapped on a C18 Acclaim PepMap100 5 µm, 0.3 µm × 5 mm cartridge (Thermo Fisher Scientific) before separation on Aurora Ultimate C18, 1.7 µm, 75 µm × 25 cm (IonOpticks). Peptides were eluted on optimized gradients of 90 min and interfaced via an EasySpray ionization source to a tribrid quadrupole mass spectrometer (Orbitrap Eclipse, Thermo Fisher Scientific) equipped with FAIMS. MS data were acquired in data-dependent mode with a Top-25 method; high resolution scans full mass scans were carried out (R = 120,000, m/z 400–1550), followed by higher energy collision dissociation with stepped collision energy range 21, 30, and 34% normalized collision energy. The tandem mass

spectra were recorded (R = 60,000, isolation window m/z 1, dynamic exclusion 50 s). Mass spectrometry measurements were cycled for 3-s durations between FAIMS CV -45 and –60 V. For data analysis, Xcalibur raw files were converted to MGF files using ProteoWizard (Chambers et al., 2012), and cross-links were analyzed by XiSearch (Mendes et al., 2019). Search conditions used 3 maximum missed cleavages with a minimum peptide length of 5. Variable modifications used were carbamidomethylation of cysteine (57.021 Da) and methionine oxidation (15.995 Da). False discovery rate was set to 5%. The cross-linking proteomics data have been deposited to the ProteomeXchange Consortium via the PRIDE partner repository (Perez-Riverol et al., 2025), with the dataset identifier PXD070863 (https://proteomecentral.proteomexchange.org/ui?pxid=PXD070863; ProteomeXchange, 2025b).

## Protein expression and purification
### Protein expression in Expi293 cells

To transfect cells, polyethyleneimine (linear MW 25,000, Polysciences) was mixed and incubated with plasmid DNA for 20 min in Opti-MEM at a mass ratio of 3:1 and added to the cells in Expi293 expression medium (A1435101; Thermo Fisher Scientific) when cells were at $3 × 10^6$/ml. Transfected cells were harvested after 48 h, washed in PBSA buffer, frozen in liquid nitrogen, and stored at –80°C until purification. Frozen cell pellets expressing GFP-Flag-CALCOCO1, 3xFlag-GFP-OPTN, and GFP-3xFlag-FHIP2A were thawed in lysis buffer (50 mM HEPES [pH 8.0], 150 mM NaCl, 1% IGEPAL CA-630 [NP40 substitute], 12% glycerol, 0.5 mM TCEP, 2 mM MgCl$_2$, and 1× cOmplete EDTA-free protease inhibitor tablets); ~50 ml lysis buffer was used per 400 ml of Expi293 cell culture and rotated for 30 min at 4°C. Lysates were then clarified at 16,000 × $g$ for 30 min, and the supernatants then incubated with anti-DYKDDDDK G1 affinity resin (L00432; GenScript) for 4 h at 4°C with agitation before washing the resin four times with lysis buffer. Resin was washed once with chaperone removal buffer (50 mM HEPES [pH 8.0], 150 mM NaCl, 0.1% Triton X-100, 0.5 mM TCEP, 50 mM MgCl$_2$, 5 µg/ml RNaseA, and 5 mM ATP) before being incubated ON at 4°C with mixing in chaperone removal buffer. Resin was subsequently washed two more times in chaperone removal buffer before being washed four times in elution buffer (50 mM HEPES [pH 8.0], 150 mM NaCl, and 0.5 mM TCEP). Proteins were eluted with 240 µg/ml Flag peptide in elution buffer, concentrated using Amicon Ultra centrifugal filters with a 100 kDa cutoff (UFC910008; Millipore), and used directly or stored at –80°C after snap freezing in liquid nitrogen.

### Expression of Rab1A fusion proteins

BL21-GOLD cells were transformed with plasmids expressing GST-Rab1-6xHis constructs (either Rab1A Q70L, Rab1A S25N, Rab1B Q67L, or Rab1B S26N) and grown ON at 37°C in a CO$_2$-free incubator. Colonies were picked and resuspended in 1 liter of LB medium at 37°C with agitation. Bacteria were grown to an OD600 of 0.7 and induced with 100 mM IPTG ON at 18°C with agitation. Cell pellets were frozen using liquid nitrogen and stored at –80°C. Frozen cell pellets were resuspended in lysis buffer (50 mM Tris-HCl, pH 7.4, 150 mM NaCl, 5 mM MgCl$_2$, 1%

Triton X-100, 5 mM β-mercaptoethanol, and 1× cOmplete EDTA-free protease inhibitor tablets) and sonicated (2 min total, with a 1-s on and 1-s off cycle at 45% amplitude) and then pelleted at 27,000 × $g$ for 30 min. Supernatants were incubated with glutathione Sepharose 4B beads (17075601; Cytiva, also written as GSH beads) for 4 h at 4°C with agitation The beads were then washed four times with lysis buffer, four times with elution buffer (50 mM HEPES [pH 8.0], 150 mM NaCl, and 0.5 mM TCEP), and eluted with elution buffer containing PreScission Protease (Cytiva, roughly 80 U of protease per 1 ml of Sepharose 4B beads slurry) ON at 4°C without agitation. Proteins were concentrated using Amicon Ultra centrifugal filter units with a 3 kDa cutoff (UFC900308; Millipore) if required. Final protein assays included either 100 mM non-hydrolyzable GTP analog (guanosine 5′-[β,γ-imido]triphosphate (Sigma-Aldrich) or 100 mM GDP as appropriate.

### Expression and purification of FHF complex
The FTS–Hook2[455–719]–FHIP2A complex was expressed and purified as described previously (Abid Ali et al., 2025). Briefly, FHF constructs were expressed using the baculovirus–Sf9 system with codon optimization. Bacmid DNA was produced by transforming DH10EmBacY cells, and DNA was isolated using a modified alkaline lysis Qiagen mini-prep protocol, as previously described (Schlager et al., 2014). P1 virus generation involved transfecting 2–3 μg of bacmid DNA (FuGene HD, Promega) into 2 ml Sf9 cell cultures (0.5 × 10⁶ cells/ml in a 6-well plate), followed by a 5–7-day incubation at 27°C without agitation. The P1 virus was collected by pipetting. For P2 virus amplification, 1 ml of P1 stock was added to 50 ml of Sf9 cells (0.5 × 10⁶ cells/ml in a 250-ml Erlenmeyer flask, a 1:50 vol/vol dilution) and incubated at 27°C with 140 rpm shaking for 72 h. P2 viral supernatant was harvested by centrifugation (4,000 × $g$, 5 min, RT). For protein expression, 5–7 ml of P2 virus was used to infect 500 ml cultures of Sf9 cells (1.5 × 10⁶ cells/ml in roller bottles). These were incubated at 27°C with 140 rpm shaking for 56 h. Cells were collected by centrifugation (4,000 × $g$, 10 min, 4°C), snap frozen in liquid nitrogen, and stored at –80°C.

Cell pellets derived from 2 liters of culture were thawed and resuspended in 45 ml of lysis buffer (50 mM HEPES, pH 7.2, 150 mM NaCl, 10% glycerol, and 1 mM DTT, supplemented with 2 mM phenylmethyl sulfonyl fluoride and two cOmplete EDTA-free protease inhibitor tablets [Roche] per 50 ml of buffer). Cells were disrupted on ice using a tight-fitting dounce homogenizer with 20–25 strokes, and the lysate was clarified at 504,000 × $g$ for 45 min at 4°C in a Ti70 rotor (Beckman Coulter). In a cold room, the cleared lysate was applied to a Bio-Rad gravity flow column containing 4 ml of pre-equilibrated Strep-Tactin Sepharose beads (2-1201-025; IBA). The flow-through was collected and reapplied once to maximize protein binding. The beads were washed with ∼450 ml of lysis buffer, followed by 200 ml of PreScission (Psc) buffer (50 mM Tris-HCl, pH 7.4, 150 mM NaCl, 1 mM EDTA, and 1 mM DTT). The beads, along with 6 ml of Psc buffer, were transferred into two 5-ml Eppendorf tubes, and 90 μl of Psc enzyme (2 mg/ml) was added to each tube. Samples were then incubated for 16 h at 4°C on a PTR-60 multirotator (Grant-Bio). The following day, the bead-elution

mixture was loaded onto a small gravity column, and residual-bound protein was eluted with 5 ml of Psc buffer. The combined eluate was concentrated using a 15 ml concentrator with a 100 kDa cutoff (UFC910008; Millipore), employing repeated 5-min spins at 4,000 × $g$ and mixing by pipetting between spins, until a concentration of ∼8 mg/ml in ∼800 μl was reached. This concentrated protein solution was clarified at 12,000 × $g$ for 5 min, and 250 μl aliquot of the supernatant was injected onto a Superose 6 10/300 column using a 500 μl sample loop, and 300 μl fractions were collected. Peak fractions were pooled and concentrated using a 4 ml 100 kDa cutoff concentrator. Glycerol was added to a final concentration of 10%, and the purified protein was stored in 5 μl aliquots. This scheme typically yielded 2–4 mg of protein.

### Expression and purification of GST-LC3B and GST-4×ubiquitin
A plasmid expressing GST-LC3B was transformed into OverExpress C41(DE3) chemically competent cells (60442; Lucigen). Plasmids for GST only or GST-4xubiquitin were transformed into BL21 *Escherichia coli*. All bacteria were grown in 2 liters 2xTY medium supplemented with 0.1 mg/ml ampicillin (AMP50; Formedium) at 37°C to an OD600 of 0.8, then expression was induced with 0.5 mM IPTG for LC3B and 0.1 mM IPTG for GST and GST-4xUb, and bacteria were incubated for a further 20 h at 16°C. Cells were pelleted at 6700 × $g$ for 15 min, frozen in liquid nitrogen, and stored at –80°C until use. LC3B pellets were resuspended in 40 ml lysis/wash buffer 1 (50 mM HEPES, pH 8.0, 500 mM NaCl, 12% glycerol, 10 mM DTT, 1 mM PMSF, and 1x EDTA-free protease inhibitor cocktail tablet [Roche]); GST and GST-4xUb pellets were resuspended in lysis buffer 2 (20 mM Tris, pH 7.4, 150 mM NaCl, 10% glycerol, protease inhibitor tablets, and 20 μg/ml DNaseI). Cells were sonicated for 5 min on ice (2 s on, 3 s off, and 60% amplitude), the lysate was cleared at 142,000 × $g$ for 15 min at 4°C, and it was filtered with a 5-μm syringe filter (Sartorius). 3 ml bed volume of glutathione Sepharose 4B beads (GE Healthcare) was added to the supernatant and rotated at 8 rpm in the cold room for 1 h. The LC3B lysate/beads mixture was transferred to a gravity flow column and washed with 300 ml lysis/wash buffer 1, then the protein was eluted with 20 ml elution buffer (50 mM HEPES, pH 8.0, 300 mM NaCl, 10 mM reduced glutathione, 10 mM DTT, 1 mM PMSF, and 1× EDTA-free protease inhibitor cocktail tablet). The eluted protein was concentrated using a 30 k cutoff concentrator (UFC903008; Millipore). The concentrated protein was subjected to gel filtration on a S75 16/60 column equilibrated with GF buffer (20 mM HEPES, pH 7.4, 300 mM NaCl, and 5 mM DTT). The peak fractions were combined and concentrated to 38 mg/ml for GST-LC3B using a 30 k cutoff concentrator. GST-LC3B was subsequently rebound to glutathione Sepharose 4B beads and washed four times with lysis 1, followed by four washes using binding buffer (50 mM HEPES [pH 8.0], 150 mM NaCl, 0.5 mM TCEP, and 5 mM MgCl₂). GST and GST-4xubiquitin lysate/beads mixtures were washed four times with lysis buffer 2, followed by four washes in binding buffer, and used immediately for downstream bead imaging assays.

### Creation of stable cell lines
Stable cell lines were generated by lentiviral transduction. To produce lentiviral particles, HEK293T cells were seeded at a

density of $8 \times 10^5$ cells per well in a 6-well plate and grown to 60–70% confluency. Cells were transfected with a mixture containing 500 ng of the packaging plasmid pMD-OGP, 500 ng of the envelope plasmid pMD-VSVG, and 1 µg of the plasmid encoding the construct of interest. DNA was initially incubated with 4.8 µl Lipofectamine 2000 according to the manufacturer's instructions (Invitrogen), diluted into 200 µl Opti-MEM, and added to cells. Forty-eight hours after transfection, the lentivirus-containing supernatant was harvested and cleared at $500 \times g$ for 3 min. For transduction, HEK293A cells were plated at 70% confluency, and the culture medium was supplemented with Polybrene to a final concentration of 8 µg/ml. The harvested viral supernatant was added to the cells at various dilutions (1:5, 1:10, 1:50, and 1:200). The plate was then sealed and centrifuged at 1,800 rpm for 2 h at RT to facilitate transduction (spinoculation). The following day, transduction efficiency was assessed by observing GFP fluorescence or by western blotting. Cell populations, which exhibited optimal fluorescence or protein expression intensity without signs of excessive viral load, were selected for further expansion.

## Mitophagy assay

Cells stably expressing the pSu9-Halo-mGFP mitophagy reporter, in addition to the indicated OPTN constructs, were further infected using lentiviral particles carrying YFP-Parkin–expressing plasmids (Addgene_23955) to ensure adequate mitophagy responses in HeLa cells. 24 h later these cells were then incubated with DMEM containing 10% FBS and 100 nm TMR HaloTag ligand (Promega) for 20 min at 37°C. Cells were washed twice before incubation with DMEM containing 10% FBS, 1 µM oligomycin, 5 µM antimycin A, and 20 µM of the pan-caspase inhibitor quinoline-Val-Asp-difluorophenoxymethylketone (HY-12305; MedChemExpress) for 24 h. Cells were then washed in ice-cold PBS, lysed in TNTE buffer (see above for composition), and insoluble debris pelleted at $16,000 \times g$, and the supernatant was separated by SDS-PAGE on NuPAGE Bis-Tris 4–12% gels. The fluorescence signal of TMR was visualized by direct imaging of the gels using the 546 nm channel of the Bio-Rad ChemiDoc MP. After imaging, gels were processed for western blotting as described above. Mitophagy flux was quantified by dividing the intensity of free Halo TMR 546 nm fluorescence by the sum of the free HALO and pSu9-Halo-mGFP TMR 546 nm fluorescence.

## In vitro Rab1-binding assays

GST-Rab1 chimeras were purified from BL21-GOLD cells as described above, but instead of eluting them from the glutathione Sepharose 4B beads, the beads were incubated with binding buffer (50 mM HEPES [pH 8.0], 150 mM NaCl, 0.5 mM TCEP, and 5 mM $MgCl_2$), with either 100 mM guanosine 5′-[β,γ-imido]triphosphate (G0635; Sigma-Aldrich) or 100 mM GDP (G7127; Sigma-Aldrich), and containing 1 µM of specified purified proteins (purified as described above). When probing total lysate, beads were incubated with total HEK293A lysate containing either 100 mM guanosine 5′-[β,γ-imido]triphosphate or 100 mM GDP and incubated ON at 4°C with rotation. Beads were then washed four times with binding buffer for purified proteins or with lysis buffer for total lysate. Beads incubated with purified proteins were eluted with binding buffer containing PreScission Protease (27084301; Cytiva) ON at 4°C without agitation, and beads incubated with HEK293A lysate were eluted with 2× Laemmli sample buffer (161–0747; Bio-Rad, the 4× stock buffer was supplemented with 10% β-mercaptoethanol and then diluted to obtain a 2× working solution) at 65°C for 5 min.

## Binding assays using GUVs

The generation of GUVs and their immobilization and visualization in an observation chamber were essentially as described previously (Tremel et al., 2021). In summary, to generate GUVs, glass coverslips were coated with 5% polyvinyl alcohol (PVA, 8148940101; Sigma-Aldrich), with excess PVA removed by spinning in a benchtop centrifuge at $1000 \times g$. PVA-coated coverslips were then dried at 60°C for 20 min. A 15 µl aliquot of the 1 mg/ml GUV lipid mixture in chloroform (18% PI liver, 10% DOPS, 7% DOPE, 55% DOPC, 10% NiNTA, 0.03% DSPE-PEG-Biotinyl, and 0.03% Lissamine Rhodamine DOPE, all lipids were obtained from Avanti Polar Lipids) was placed onto the PVA-coated side of a glass coverslip by twice carefully spreading 7.5 µl of lipid mixture over the entire surface of the coverslip using a pipette tip. Coverslips were then dried in a vacuum desiccator for 2 h and then placed in a well of a 24-well plate and carefully covered in 220 µl of filtered swelling solution (0.5 M sucrose) for 1–2 h at RT to induce GUV production. GUVs were then removed from the well using a cutoff pipette tip and transferred to a 1.5-ml Eppendorf tube, which had been coated using 5 mg/ml BSA (A7030; Sigma-Aldrich) for 1 h with agitation and then rinsed once with the swelling solution. To immobilize GUVs, wells of an 8-well glass bottom slide (80827; ibidi) were incubated for 15 min with 100 µl avidin egg white (A2667; Life Technologies) at 0.1 mg/ml PBS, supplemented with 1 mg/ml BSA for 15 min, and then washed two times with observation buffer (25 mM HEPES, pH 8.0, and 271.4 mM NaCl). Observation buffer was then added to the wells, followed by addition of 48 µl of GUVs. 10 µM of His-tagged Rab1 proteins (fourfold molar excess over NiNTA lipids) were added to the immobilized GUVs and incubated at RT for 30 min. Unbound Rab1 was removed by carefully adding and removing 360 µl of observation buffer four times before adding observation buffer containing 1 µM of the desired protein. The final protein solution included either 100 mM non-hydrolyzable GTP analog (guanosine 5′-[β,γ-imido]triphosphate) or 100 mM GDP, as appropriate. For assays using beads, washed glutathione Sepharose 4B beads saturated with proteins of interest (either GST-LC3B, GST-4xUbiquitin, or GST control) were added to a 384-well plate (781856; Greiner) and were incubated with binding buffer (50 mM HEPES [pH 8.0], 150 mM NaCl, 0.5 mM TCEP, and 5 mM $MgCl_2$) containing 1 µM of the protein of interest. GUVs or beads were imaged using a Zeiss LSM 880 Confocal microscope running Zeiss ZEN imaging software with either a 40× oil-immersion objective (1.3 NA, Airyscan mode) for GUVs or a 10× air objective (0.30 NA) for beads at RT. Images were analyzed with Fiji (https://fiji.sc) using a custom plugin. Briefly, a subsection of the GUV membrane or surface of the bead was selected with the selection tool using the lissamine-rhodamine

(GUV) or DIC (beads) channel, and then the mean fluorescence intensity of the GFP signal was quantified.

### Light microscopy
Cells cultured on coverslips were fixed for 15 min in 4% formaldehyde in PBS and then permeabilized for 5 min at RT with a 50 µg/ml digitonin in PBS. Coverslips were then washed with PBS and blocked for 20 min using 5% BSA (Roche) in PBS. Incubation with primary antibodies, diluted in 5% BSA in PBS, was performed for 1 h at RT with coverslips carefully placed onto 50 µl drops of antibody-containing solution. After washing, coverslips were incubated with secondary antibodies, also diluted in 5% BSA, for 1 h at RT. After final washes with PBS and deionized water, coverslips were mounted in ProLong Gold Antifade (P36930; Thermo Fisher Scientific) on a glass slide ON at RT. Image acquisition was on a Zeiss LSM 880 Airyscan confocal microscope running Zeiss ZEN imaging software, using a 40× oil-immersion objective (1.3 NA, Airyscan mode) at RT.

### Statistics
Statistical analyses were conducted with GraphPad Prism v10 (RRID; SCR_002798), with specific methodologies outlined within each figure legend. An assumption of data normality was made based on assay characteristics and visual inspection; formal verification of normality was not performed due to the limited sample sizes involved. Precise "$n$" values for each experiment are stated in the figure legends. When applicable, levels of statistical significance are indicated in the figures using asterisks (*$P < 0.05$, **$P < 0.01$, and ***$P < 0.001$). Sample sizes, which typically ranged from three to six, were determined by considering preliminary experimental results and inherent assay variability. These sample numbers are typical of published literature utilizing comparable methods, and all data that met acceptable experimental quality were included. The assignment of seeded cells to different experimental groups was carried out randomly. The helical wheel plot (Fig. 2 D) was produced using HeliQuest (https://heliquest.ipmc.cnrs.fr/index.html).

### Online supplemental material
Fig. S1 shows expression, localization, and activity of Rab1 MitoID constructs. Fig. S2 shows characterization of Rab1 binding to the FHF complex and to CALCOCO1. Fig. S3 shows mutation of the Rab1-binding site in OPTN does not affect its dimerization or binding to LC3 or ubiquitin. Table S1 shows proteomic data from Rab1 MitoID as shown in Fig. 1 and Fig. S1.

### Data availability
The data underlying all figures are available in the published article and its online supplemental material, with the mass spectrometry proteomics and cross-linking data openly available in the ProteomeXchange Consortium with the dataset identifiers PXD065220 and PXD070863.

## Acknowledgments
We thank Keith Boyle, Sami Chaaban, Thomas Mund, Saulė Spokaite, Roger Williams, and members of the Munro lab for advice and helpful discussions and Jessica Bertram for help in piloting Rab1 MitoID. Mass spectrometry was performed by the facility at the MRC LMB under the guidance of Catarina Franco.

This work was supported by the Medical Research Council, as part of United Kingdom Research and Innovation (also known as UK Research and Innovation) file reference number MC_U105178783. Open Access funding provided by MRC Laboratory of Molecular Biology.

Author contributions: Alexander R. van Vliet: conceptualization, data curation, formal analysis, investigation, methodology, project administration, resources, validation, visualization, and writing—original draft, review, and editing. Alison K. Gillingham: investigation and resources. Tomos E. Morgan: investigation. Yohei Ohashi: methodology and resources. Tom S. Smith: formal analysis. Ferdos Abid Ali: resources and writing—review and editing. Sean Munro: conceptualization, funding acquisition, supervision, and writing—review and editing.

Disclosures: The authors declare no competing interests exist.

Submitted: 13 July 2025

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

# Supplemental material

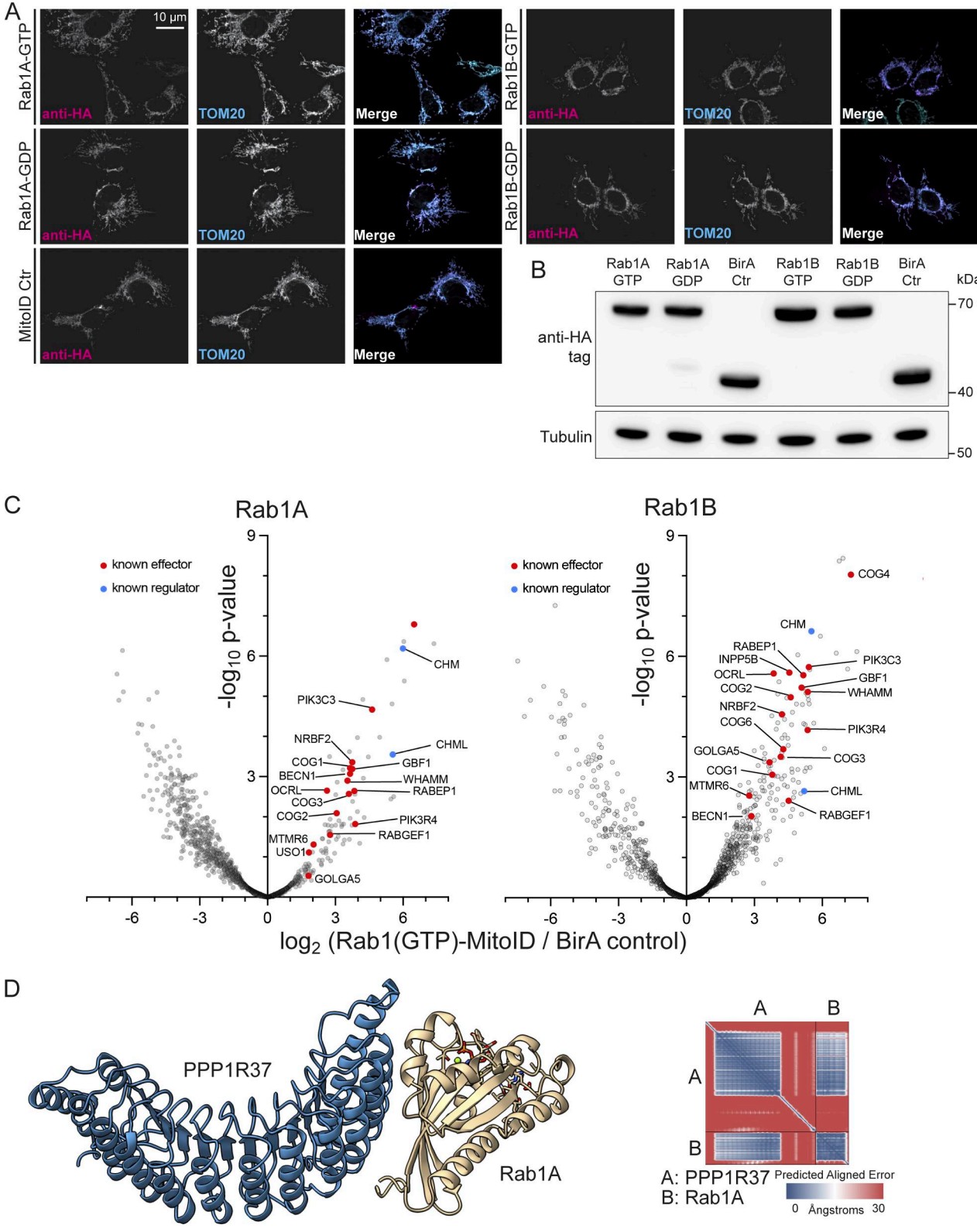

Figure S1. **Expression, localization, and activity of Rab1 MitoID constructs. (A)** Confocal images of HEK293A cells expressing the indicated Rab1 MitoID constructs, in addition to the BirA control construct. Cells were stained for the HA tag in the MitoID constructs and the mitochondrial protein Tom20. Both Rab1A and Rab1B contain mutations that lock them in a either a GTP- or GDP-bound form (QL or SN, respectively). **(B)** Representative immunoblot showing comparable expression levels of all MitoID constructs used in the proteomics study. **(C)** Volcano plots comparing the abundances of biotinylated proteins obtained with MitoID of GTP-locked Rab1A or Rab1B (Rab1A Q70L; Rab1B Q67L) vs BirA alone negative control. Rab1 itself is not shown, and known effectors and regulators are indicated. **(D)** Structure of the Rab1A:GTP:Mg$^{2+}$:PPP1R37 complex as predicted by AlphaFold 3, with accompanying PAE plot. Source data are available for this figure: SourceData FS1.

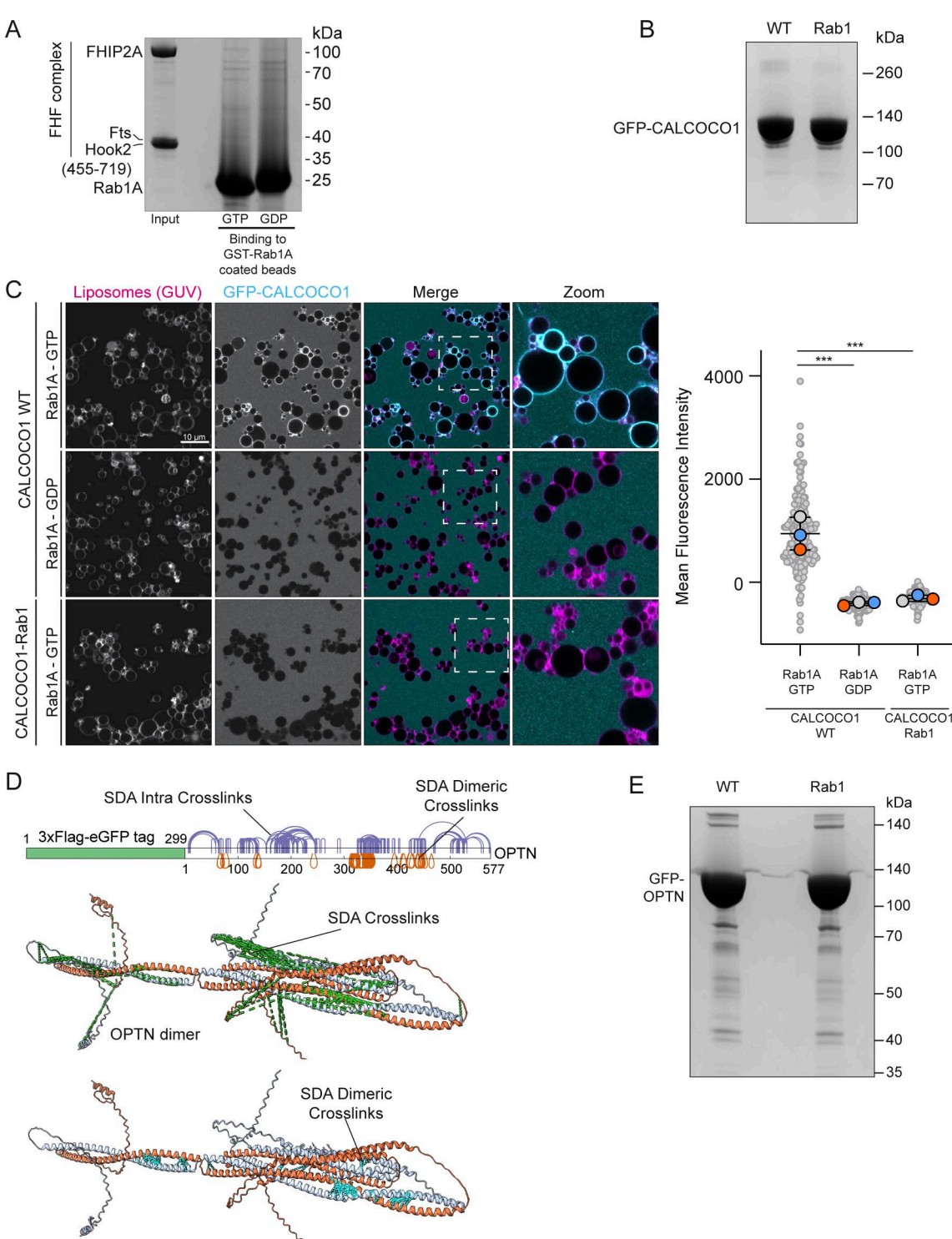

Figure S2. **Characterization of Rab1 binding to the FHF complex and CALCOCO1. (A)** Coomassie gel showing an in vitro–binding assay using beads coated with GST-Rab1A and purified FHF complex containing FHIP2A. Both GTP- and GDP-locked Rab1A proteins were used as indicated. **(B)** Representative Coomassie blue–stained gel showing expression of either WT or Rab1-binding mutant GFP-CALCOCO1. Proteins were purified from the same amount of starting material, and equal protein amounts were loaded on the gel. **(C)** GUV-binding assay using GTP- or GDP-locked Rab1A bound to GUVs before applying GFP-CALCOCO1 WT or Rab1-binding mutant. Each large (colored) datapoint in the graph depicts the average mean fluorescence intensity of GFP-CALCOCO1 WT or Rab1-binding mutant on a selection of GUV membrane and represents an independent experiment (*n* = 3), with smaller gray datapoints representing all the technical replicates (AU, arbitrary units). The mean ± SD is indicated. \*\*\*P < 0.001 (one-way ANOVA with Tukey's multiple comparisons test). **(D)** Sulfo-NHS-diazarine (SDA) intra-protein and dimer cross-links (purple and orange lines) mapped onto the AlphaFold 3 predicted structure of the OPTN homodimer. Intra-protein cross-links mapped onto the structure are pictured as a green dashed line; dimer cross-links are pictured as a cyan dashed line. **(E)** Representative Coomassie blue–stained gel showing expression of either WT or Rab1-binding mutant GFP-OPTN. Proteins were purified from the same amount of starting material, and equal protein amounts were loaded on the gel. Source data are available for this figure: SourceData FS2.

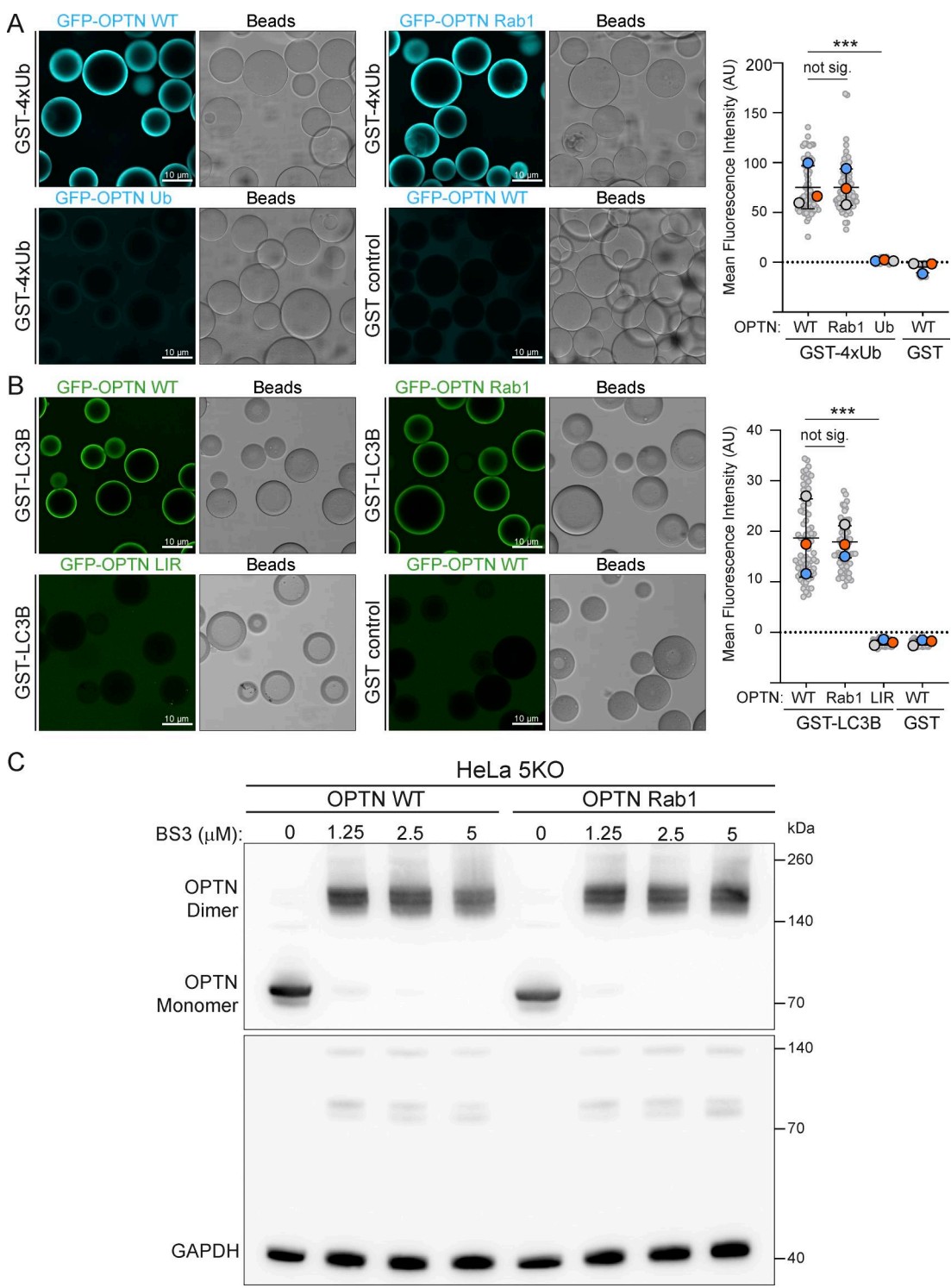

Figure S3. **Mutation of the Rab1-binding site in OPTN does not affect its dimerization or binding to LC3 or ubiquitin. (A)** Micrographs of beads coated with GST-4×ubiquitin (GST-4xUb) and incubated with either 1 μM GFP-OPTN (WT) or versions with mutations in the Rab1 or ubiquitin-binding sites. Each large datapoint in the bar graph depicts the average mean fluorescence intensity on a selection of beads and represents an independent experiment (n = 3), with smaller gray datapoints representing all the technical replicates (AU, arbitrary units). The mean ± SD is indicated. ***P < 0.001 (one-way ANOVA with Tukey's multiple comparisons test). **(B)** Micrographs of beads coated with GST-LC3B and incubated with either 1 μM GFP-OPTN (WT) or versions with mutations in the Rab1 or LC3 (LIR)-binding sites. Each large datapoint in the bar graph depicts the average mean fluorescence intensity on a selection of beads and represents an independent experiment (n = 3), with smaller gray datapoints representing all the technical replicates (AU, arbitrary units). The mean ± SD is indicated. ***P < 0.001 (one-way ANOVA with Tukey's multiple comparisons test). **(C)** Immunoblot of cell lysates of pentaKO HeLa cells stably expressing WT OPTN or the OPTN Rab1-binding mutant (representative of three repeats). Lysates were incubated with increasing concentrations of the cross-linker bis(sulfosuccinimi-dyl)suberate (BS3) at RT and quenched at 50 mM Tris-HCl prior to SDS-PAGE and blotting. With cross-linker, OPTN dimers are readily detected. GAPDH is a loading control and a negative control as it does not readily dimerize. Source data are available for this figure: SourceData FS3.

**Provided online is Table S1. Table S1 shows proteomic data from Rab1 MitoID as shown in Fig. 1 and Fig. S1.**

