## [Peer Review File · The Journal of Cell Biology]

A Rab1 interactome illuminates a dual role in autophagy and membrane trafficking

Alexander van Vliet, Alison Gillingham, Tomos Morgan, Yohei Ohashi, Tom Smith, Ferdos Abid Ali, and Sean Munro

Corresponding Author(s): Sean Munro, MRC Laboratory of Molecular Biology and Alexander van Vliet, MRC Laboratory of Molecular Biology

Review Timeline:

Submission Date:	2025-07-13
Editorial Decision:	2025-09-02
Revision Received:	2025-11-20
Editorial Decision:	2025-11-21
Revision Received:	2025-12-01

Monitoring Editor: Christian Ungermann

Scientific Editor: Andrea Marat

Transaction Report:

DOI: <https://doi.org/10.1083/jcb.202507084>

September 2, 2025

Re: JCB manuscript #202507084

Sean Munro
MRC Laboratory of Molecular Biology

Dear Sean,

Thank you for submitting your manuscript entitled "A Rab1 interactome illuminates a dual role in autophagy and membrane trafficking". The manuscript was assessed by expert reviewers, whose comments are appended to this letter. We invite you to submit a revision if you can address the reviewers' key concerns, as outlined here.

You will see that the reviewers appreciate that your study provides a useful resource for the field along with new insights into the role of Rab1 in trafficking and autophagy that should be interesting for the JCB readership. While overall they find that the data is high-quality, they have provided constructive suggestions which we hope you agree will further improve your study and that you will be able to address in a revised study. In particular, regarding the questions from reviewer 2 about the OPTN mutation that affects Rab1 binding we agree that Atg8 and cargo interaction should be tested as an important control. Likewise, the OPTN mutant should be tested for self-interaction (point 3). The last point of reviewer 2 is experimentally beyond the scope of the study, though should be addressed in the response letter.

GENERAL GUIDELINES:

Text limits: Character count for a Report is < 20,000, not including spaces. Count includes title page, abstract, introduction, the joint Results & Discussion, and acknowledgments. Count does not include materials and methods, figure legends, references, tables, or supplemental legends.

Figures: Reports may have up to 5 main text figures. To avoid delays in production, figures must be prepared according to the policies outlined in our Instructions to Authors, under Data Presentation, <https://jcb.rupress.org/site/misc/ifora.xhtml>. All figures in accepted manuscripts will be screened prior to publication.

IMPORTANT: It is JCB policy that if requested, original data images must be made available. Failure to provide original images upon request will result in unavoidable delays in publication. Please ensure that you have access to all original microscopy and blot data images before submitting your revision.

Supplemental information: There are strict limits on the allowable amount of supplemental data. Reports may have up to 3 supplemental figures. Up to 10 supplemental videos or flash animations are allowed. A summary of all supplemental material should appear at the end of the Materials and methods section.

Please note that JCB now requires authors to submit Source Data used to generate figures containing gels and Western blots with all revised manuscripts. This Source Data consists of fully uncropped and unprocessed images for each gel/blot displayed in the main and supplemental figures. For assays performed using capillary electrophoresis and/or immunoassay-based detection, authors should instead provide the electropherogram graph(s) for each experiment, plotting fluorescence/chemiluminescence intensity vs. molecular weight/size. Please be sure to provide one Source Data file for each figure gels, blots, and/or capillary electrophoresis assays along with your revised manuscript files. File names for Source Data figures should be alphanumeric without any spaces or special characters (i.e., SourceDataF#, where F# refers to the associated main figure number or SourceDataFS# for those associated with Supplementary figures). For traditional gels and blots, the lanes of the gels/blots should be labeled as they are in the associated figure, the place where cropping was applied should be marked (with a box), and molecular weight/size standards should be labeled wherever possible. For capillary electrophoresis assays, each trace in the graph should be color-coded and labeled to indicate which protein, gene, or sample is being measured (please try to avoid red/green combinations to accommodate our color-blind readers).

The typical timeframe for revisions is three to four months. If you anticipate any difficulties in meeting this aforementioned

revision time limit, please contact us and we can work with you to find an appropriate time frame for resubmission. Please note that papers are generally considered through only one revision cycle, so any revised manuscript will likely be either accepted or rejected.

Thank you for this interesting contribution to Journal of Cell Biology. You can contact us at the journal office with any questions at cellbio@rockefeller.edu.

Sincerely,

Christian Ungermann
Monitoring Editor

Andrea L. Marat
Deputy Editor

Journal of Cell Biology

Reviewer #1 (Comments to the Authors (Required)):

In this study the authors use 'MitolD', a proximity proteomics approach developed in the Munro lab, to characterize the Rab1A and Rab1B interactomes. In addition to expected interactors, they identified several novel interactors. The authors performed a more detailed characterization of three hits, leading to important mechanistic findings.

The authors first characterize FHIP2A, a known dynein adaptor, as a Rab1 effector. Previously FHIP2A was identified as a putative Rab1 effector but direct binding was weak. Here the authors find that strong binding requires the presence of liposome membranes. They also identify an amphipathic helix in FHIP2A that is likely a membrane binding element and they determine that this helix is required for interaction with Rab1 on liposomes.

The authors then characterize the selective autophagy receptors OPTN and CALCOCO1 as Rab1 effectors. They observe robust, GTP-dependent interactions with Rab1 in the presence and absence of liposome membranes. They find that cross-linking mass spectrometry results are consistent with the AlphaFold-predicted structures of Rab1-OPTN and Rab1-CALCOCO1 complexes. The authors used this information to design a mutant version of OPTN that does not interact with Rab1, and they found that this mutant protein was not capable of supporting mitophagy in cells.

Overall I think this is a nice study that not only provides a useful resource to the field but also makes two important findings regarding specific roles of Rab1 in trafficking and autophagy. One can always ask for more experiments to dive deeper but I am instead happy to recommend publication of this important work in its current form with only a few minor suggestions:

1. The authors might reconsider their description of Rab1 as a "molecular glue", as this terminology has been used primarily for small molecules (not proteins) that mediate protein-protein interactions.
2. In the discussion the authors suggest a possible handoff mechanism in which OPTN would first bind Rab1 and then bind to LC3. Is there a reason why these two interactions need to be sequential, rather than simultaneous (i.e. "coincidence detection").
3. Previous studies from the Ferro-Novick lab reported that Rab1/Ypt1 recruits Atg1 and the kinase CK1delta. The authors might consider discussing whether the homologous proteins were detected in their MitolD dataset.

Reviewer #2 (Comments to the Authors (Required)):

In the manuscript, Vliet et al., used MitolD, a modified proximity biotinylation approach, to identify new interactors of Rab1A and Rab1B. This approach is based on a publication by Gillingham et al. (2019). It identifies unrecognised interactors of Rab1A and Rab1B in a GTP-dependent manner. The authors focused their analysis on FHIP2A, which is a cargo adaptor for dynein and the selective autophagy receptors OPTN and CALCOCO1.

Overall, the study is well written, and the data it provides is of high quality. I congratulate the authors on this, and I have no

major concerns in this regard. However, as the method and some of the characterised hits (e.g. OPTN - PMID: 28843006) have been published before, the study's novelty is somewhat limited. While I acknowledge that the authors have provided higher-quality data on the interaction between these proteins in vitro and in vivo, I feel that the publication is somewhat preliminary in its current state. The following points would strengthen the manuscript:

Major points:

- Is there evidence in this or their earlier work that the autophagy receptors bind other Rab proteins, and if so, is this mechanism common? If not, what is the rationale behind the targeting of Rab1 by OPTN and CALCOCO1 in particular?
- Does the OPTN mutation that affects Rab1 binding also impair the interaction with Atg8? Is cargo binding similarly influenced?
- The CC domain in receptors such as OPTN is essential for self-oligomerisation. It is possible that impaired self-oligomerisation leads to a decrease in mitophagy that is not directly linked to Rab1 binding. The authors should provide evidence that self-oligomerisation is unaffected.
- The interaction between OPTN and Rab1, and the functional consequences of this interaction, are very interesting. Typically, OPTN is targeted to the phagophore via binding to LC3/GABARAP. In the discussion, the authors suggest that Rab1 binding may be an early mechanism by which receptors bind to the initial phagophore to hold the cargo in place. This is intriguing. Supporting this theory with experimental data would address a key aspect currently lacking in the study: why early OPTN function is required.

Revisions in Response to Reviewers' Comments.

We are very grateful indeed to the reviewers for their positive comments about our work and their constructive suggestions for improvements. We have followed these suggestions, as described below, and have also made some minor changes to ensure that the manuscript conforms to the JCB guidelines.

Reviewer #1 (Comments to the Authors):

In this study the authors use 'MitolD', a proximity proteomics approach developed in the Munro lab, to characterize the Rab1A and Rab1B interactomes. In addition to expected interactors, they identified several novel interactors. The authors performed a more detailed characterization of three hits, leading to important mechanistic findings.

The authors first characterize FHIP2A, a known dynein adaptor, as a Rab1 effector. Previously FHIP2A was identified as a putative Rab1 effector but direct binding was weak. Here the authors find that strong binding requires the presence of liposome membranes. They also identify an amphipathic helix in FHIP2A that is likely a membrane binding element and they determine that this helix is required for interaction with Rab1 on liposomes.

The authors then characterize the selective autophagy receptors OPTN and CALCOCO1 as Rab1 effectors. They observe robust, GTP-dependent interactions with Rab1 in the presence and absence of liposome membranes. They find that cross-linking mass spectrometry results are consistent with the AlphaFold-predicted structures of Rab1-OPTN and Rab1-CALCO1 complexes. The authors used this information to design a mutant version of OPTN that does not interact with Rab1, and they found that this mutant protein was not capable of supporting mitophagy in cells.

Overall I think this is a nice study that not only provides a useful resource to the field but also makes two important findings regarding specific roles of Rab1 in trafficking and autophagy. One can always ask for more experiments to dive deeper but I am instead happy to recommend publication of this important work in its current form with only a few minor suggestions:

1. The authors might reconsider their description of Rab1 as a "molecular glue", as this terminology has been used primarily for small molecules (not proteins) that mediate protein-protein interactions.

We agree that this terminology could cause confusion, and we have replaced the term with "molecular landmark".

2. In the discussion the authors suggest a possible handoff mechanism in which OPTN would first bind Rab1 and then bind to LC3. Is there a reason why these two interactions need to be sequential, rather than simultaneous (i.e. "coincidence detection").

This is a good point that gets to the heart of our thinking about the role of Rab1. In our hypothesis, the Rab1-OPTN binding happens very early in the autophagy initiation pathway whilst the VPS34 complex is also being recruited by Rab1 to start the production of PI3P. This PI3P will go on to recruit WIPI2 which will lead to the recruitment of the machinery to lipidate LC3. Rab1 would thus provide the initial binding to the autophagy receptors, at a time when the levels of PI3P, and hence lipidated LC3, are still low. Once autophagy initiation has progressed further and lipidated LC3 has accumulated, the latter can also bind OPTN and take over some or all of the anchoring role from Rab1.

3. Previous studies from the Ferro-Novick lab reported that Rab1/Ypt1 recruits Atg1 and the kinase CK1delta. The authors might consider discussing whether the homologous proteins were detected in their MitolD dataset.

We have added a brief discussion of this point to the Discussion section:

"In yeast, the Rab1 orthologue Ypt1 has been reported to bind to the kinases Atg1 and Hrr25, but our MitolD approach did not identify their mammalian orthologues ULK1 and CSNK1D as hits and so was not informative of whether or not they are Rab1 effectors (Wang et al., 2015; Wang et al., 2013)."

Reviewer #2 (Comments to the Authors):

In the manuscript, Vliet et al., used MitolD, a modified proximity biotinylation approach, to identify new interactors of Rab1A and Rab1B. This approach is based on a publication by Gillingham et al. (2019).

It identifies unrecognised interactors of Rab1A and Rab1B in a GTP-dependent manner. The authors focused their analysis on FHIP2A, which is a cargo adaptor for dynein and the selective autophagy receptors OPTN and CALCOCO1.

Overall, the study is well written, and the data it provides is of high quality. I congratulate the authors on this, and I have no major concerns in this regard. However, as the method and some of the characterised hits (e.g. OPTN - PMID: 28843006) have been published before, the study's novelty is somewhat limited.

We thank the reviewer for their kind words on the quality of the data in the manuscript. The Rab1-OPTN link is only part of our findings, with the paper also reporting several strong hits that are novel of which FHIP2A is also characterised in depth and will be of interest to the molecular motor field. We acknowledge that a Rab1-OPTN link had been reported before but, as we argue in the text and as this reviewer agrees, we believe we have provided extensive and robust molecular data to support this interaction and have also provided clear evidence of its *in vivo* significance. To further augment the novelty of our study we have, in addition to the changes described below, also added experimental evidence to show that mutating the Rab1 binding site we identified for CALCOCO1 using structural prediction (Figure 4) does indeed abolish Rab1 binding *in vitro* (new data shown in Figure S2C).

While I acknowledge that the authors have provided higher-quality data on the interaction between these proteins *in vitro* and *in vivo*, I feel that the publication is somewhat preliminary in its current state. The following points would strengthen the manuscript:

Major points:

- Is there evidence in this or their earlier work that the autophagy receptors bind other Rab proteins, and if so, is this mechanism common? If not, what is the rationale behind the targeting of Rab1 by OPTN and CALCOCO1 in particular?

As we note in the results, OPTN has been proposed to act in membrane traffic as well as autophagy. The membrane traffic role is thought to reflect an interaction with the closely related Rabs, Rab8A, Rab8B and Rab10. However, it has been reported that simultaneously knocking out all three of these Rabs has no effect on autophagy, whilst Rab1 is the only Rab that has been found to be required in the early steps of autophagy. The role of CALCOCO1 in other processes is less well characterised, but given the apparent unique requirement for Rab1 in early steps of autophagy it again seems likely that it is its binding to Rab1 that is relevant to autophagy. To clarify this, we have added extra text to the Discussion as follows:

“OPTN also binds to the related GTPases Rab8A, Rab8B and Rab10 in its role in trafficking, but removal of all three does not affect autophagy and so it seems likely that it is the binding to Rab1 that is relevant to this role (Okatsu et al 2025; Zhang et al 2024).”

- Does the OPTN mutation that affects Rab1 binding also impair the interaction with Atg8? Is cargo binding similarly influenced?

This is an excellent point. While the ubiquitin (cargo) binding domain in OPTN is quite distal from the Rab1 binding site we identified, the LC3 interacting region (LIR) is adjacent to the site. In the revised manuscript we have now confirmed that the Rab1 binding mutant that we generated has no detectable effect on OPTN's capacity to bind either ubiquitin or LC3B. This was demonstrated using purified proteins, and as a positive control we also tested forms of OPTN with mutations in the ubiquitin or LC3 binding sites and found that for these we could readily detect the loss of binding (see new figure, Fig. S3A and B).

- The CC domain in receptors such as OPTN is essential for self-oligomerisation. It is possible that impaired self-oligomerisation leads to a decrease in mitophagy that is not directly linked to Rab1 binding. The authors should provide evidence that self-oligomerisation is unaffected.

This is another good point, and we have now included an experiment where we utilised chemical crosslinking to detect the OPTN dimer and find that the mutation in the Rab1 binding has no detectable effect on OPTN dimerisation (see new figure, Fig. S3C).

- The interaction between OPTN and Rab1, and the functional consequences of this interaction, are very interesting. Typically, OPTN is targeted to the phagophore via binding to LC3/GABARAP. In the discussion, the authors suggest that Rab1 binding may be an early mechanism by which receptors bind to the initial phagophore to hold the cargo in place. This is intriguing. Supporting this theory with experimental data would address a key aspect currently lacking in the study: why early OPTN function is required.

This is a very interesting question. Our work and that of others indicates that Rab1 is recruited very early during autophagosome formation, in part to direct production of PI3P by the VPS34 complex I. An additional role for Rab1 in recruiting receptors at this early stage would presumably accelerate the process of cargo recognition and hence the efficiency and speed of their destruction by autophagy. Given that cargo receptors function to remove damaged cellular structures or invading pathogens, then speed would seem of an essence. To add further weight to our findings we have added new data in which we compare the effect on OPTN activity of mutating the Rab1 binding site to that of mutating the bindings sites for LC3/GABARAP or of cargo (ubiquitin) (see new panels Fig 5C and D). We find that whilst loss of ubiquitin binding all but abolishes OPTN activity, mutation of the Rab1 or LC3 bindings sites has a significant but only partial effect, indicating that both interactions play a role. Analysing the precise timing and purpose of each interaction will of course take extensive live imaging and kinetic analysis using a range of cargos and receptor mutants and so would seem beyond the scope of this initial identification of Rab1 effectors. However, we feel that our work provides robust data that Rab1 is a player in cargo recognition and hence of clear relevance to this important aspect of autophagy.

November 21, 2025

RE: JCB Manuscript #202507084R

Sean Munro
MRC Laboratory of Molecular Biology

Dear Sean,

Thank you for submitting your revised manuscript entitled "A Rab1 interactome illuminates a dual role in autophagy and membrane trafficking". We appreciate your thorough revisions and changes to the text and would therefore be happy to publish your paper in JCB pending final revisions necessary to meet our formatting guidelines (see details below).

A. MANUSCRIPT ORGANIZATION AND FORMATTING:

- 1) Text limits: Character count for Reports is < 20,000, not including spaces. Count includes abstract, introduction, ** combined results and discussion, and acknowledgments. Count does not include title page, figure legends, materials and methods, references, tables, or supplemental legends.
- 2) Figures limits: Reports may have up to 5 main text figures.
- 3) Figure formatting: Scale bars must be present on all microscopy images, including inset magnifications. Molecular weight or nucleic acid size markers must be included on all gel electrophoresis. Aspect ratios of images may not be altered.
- 4) Statistical analysis: Error bars on graphic representations of numerical data must be clearly described in the figure legend. The number of independent data points (n) represented in a graph must be indicated in the legend. Statistical methods should be explained in full in the materials and methods. For figures presenting pooled data the statistical measure should be defined in the figure legends. Please also be sure to indicate the statistical tests used in each of your experiments (either in the figure legend itself or in a separate methods section) as well as the parameters of the test (for example, if you ran a t-test, please indicate if it was one- or two-sided, etc.). Also, if you used parametric tests, please indicate if the data distribution was tested for normality (and if so, how). If not, you must state something to the effect that "Data distribution was assumed to be normal but this was not formally tested."
- 5) Abstract and title: The abstract should be no longer than 160 words and should communicate the significance of the paper for a general audience. The title should be less than 100 characters including spaces. Make the title concise but accessible to a general readership.
- 6) Materials and methods: Should be comprehensive and not simply reference a previous publication for details on how an experiment was performed. Please provide full descriptions in the text for readers who may not have access to referenced manuscripts.
- 7) All antibodies, cell lines, animals, and tools used in the manuscript should be described in full, including accession numbers for materials available in a public repository such as the Resource Identification Portal. Please be sure to provide the sequences for all of your primers/oligos and RNAi constructs in the materials and methods. You must also indicate in the methods the source, species, and catalog numbers (where appropriate) for all of your antibodies. Please also indicate the acquisition and quantification methods for immunoblotting/western blots.
- 8) Microscope image acquisition: The following information must be provided about the acquisition and processing of images:
 - a. Make and model of microscope
 - b. Type, magnification, and numerical aperture of the objective lenses
 - c. Temperature
 - d. Imaging medium
 - e. Fluorochromes
 - f. Camera make and model
 - g. Acquisition software
 - h. Any software used for image processing subsequent to data acquisition. Please include details and types of operations involved (e.g., type of deconvolution, 3D reconstitutions, surface or volume rendering, gamma adjustments, etc.).

10) Supplemental materials: There are strict limits on the allowable amount of supplemental data. Reports may have up to 3 supplemental figures. Please also note that tables, like figures, should be provided as individual, editable files. A summary of all supplemental material should appear at the end of the Materials and methods section.

13) ORCID IDs: ORCID IDs are unique identifiers allowing researchers to create a record of their various scholarly contributions in a single place. Please note that ORCID IDs are now *required* for all authors. At resubmission of your final files, please be sure to provide your ORCID ID and those of all co-authors.

Please note that JCB now requires authors to submit Source Data used to generate figures containing gels and Western blots with all revised manuscripts. This Source Data consists of fully uncropped and unprocessed images for each gel/blot displayed in the main and supplemental figures. For assays performed using capillary electrophoresis and/or immunoassay-based detection, authors should instead provide the electropherogram graph(s) for each experiment, plotting fluorescence/chemiluminescence intensity vs. molecular weight/size. Please be sure to provide one Source Data file for each figure gels, blots, and/or capillary electrophoresis assays along with your revised manuscript files. File names for Source Data figures should be alphanumeric without any spaces or special characters (i.e., SourceDataF#, where F# refers to the associated main figure number or SourceDataFS# for those associated with Supplementary figures). For traditional gels and blots, the lanes of the gels/blots should be labeled as they are in the associated figure, the place where cropping was applied should be marked (with a box), and molecular weight/size standards should be labeled wherever possible. For capillary electrophoresis assays, each trace in the graph should be color-coded and labeled to indicate which protein, gene, or sample is being measured (please try to avoid red/green combinations to accommodate our color-blind readers).

Journal of Cell Biology now requires a data availability statement for all research article submissions. These statements will be published in the article directly above the Acknowledgments. The statement should address all data underlying the research presented in the manuscript. Please visit the JCB instructions for authors for guidelines and examples of statements at (<https://rupress.org/jcb/pages/editorial-policies#data-availability-statement>).

B. FINAL FILES:

****It is JCB policy that if requested, original data images must be made available to the editors. Failure to provide original images upon request will result in unavoidable delays in publication. Please ensure that you have access to all original data images prior to final submission.****

****The license to publish form must be signed before your manuscript can be sent to production. A link to the electronic license to publish form will be sent to the corresponding author only. Please take a moment to check your funder requirements before choosing the appropriate license.****

Thank you for your attention to these final processing requirements. Please revise and format the manuscript and upload materials within 7 days. If you need an extension for whatever reason, please let us know and we can work with you to determine a suitable revision period.

Thank you for this interesting contribution, we look forward to publishing your paper in Journal of Cell Biology.

Sincerely,

Christian Ungermann
Monitoring Editor

Andrea L. Marat
Deputy Editor

Journal of Cell Biology